# Mitotic microhomology-mediated break-induced replication promotes chromoanasynthesis

Greg H. P. Ngo [1] ✉, Kez Cleal [1], Sara Seifan [1], Vanda Miklos[1], Szymon A. Barwacz [2], Brian L. Ruis[3], Siamak A. Kamranvar[1], Julia W. Grimstead[1], Ying Liu [2], Eric A. Hendrickson[3] & Duncan M. Baird [1] ✉

Chromoanasynthesis is a form of complex chromosomal rearrangement (CCR) commonly detected in cancers and congenital disorders, but the mechanism underlying its generation remain elusive. Here we develop a single-molecule long-read DNA sequencing approach to characterise ultra-complex mutational events, consistent with chromoanasynthesis, occurring at shortened telomeres and sub-telomeric DNA double-strand breaks in human cells. Our data reveal that chromoanasynthesis is generated by microhomology-mediated break-induced replication (MM-BIR), occurring specifically in mitosis. Surprisingly, this mitotic pathway involves a collaboration between microhomology-mediated end-joining (MMEJ) and BIR, where MMEJ proteins initiate a Polδ-dependent BIR pathway that is regulated by PIF1, POLD3 and PCNA. This pathway is highly prone to template switching and can generate dramatic amplification of genomic loci in a single event. Our findings help explain the extreme mutagenic nature of chromoanasynthesis and establish mitotic MM-BIR as a key driver of CCRs, with important implications for the origin of cancers and congenital disorders.

Complex chromosomal rearrangements (CCRs) are mutational events involving large numbers of clustered chromosomal rearrangements that are frequently observed in cancer and congenital disorders[1]. The best characterised subpathway of CCR is chromothripsis, which is generated by the random joining of shattered chromosomal DNA[2]. Previous studies have identified two main drivers of chromosome shattering: the formation of chromatin bridges and micronuclei, both of which occur due to errors in chromosomal segregation[3–7]. Chromothripsis is detected in a majority of cancers and is considered to be important in driving disease progression[8,9]. Chromoanasynthesis is another form of catastrophic CCR that appears to have a distinct origin from chromothripsis[10]. Chromoanasynthesis involves large numbers of templated insertion events with frequent template switching, which results in increased copy number and often contains tightly clustered

breakpoints with microhomologies at the junctions[10–12]. These mutational signatures are not compatible with an 'end-joining of fragmented DNA' model as proposed for chromothripsis, but chromoanasynthesis can occur alongside chromothripsis in cancers or cell models[5,8,13–15].

Chromoanasynthesis is commonly detected in cancers and congenital disorders[1,12]. This catastrophic process promotes cancer progression by amplifying oncogenes[15–17], but the mechanism underlying its usage remains unclear. One current model proposes the involvement of the microhomology-mediated break-induced replication (MM-BIR) pathway, which involves multiple rounds of strand invasion, DNA end annealing to microhomologies, initiation of DNA replication and dissociation, leading to copying of DNA sequence from multiple genomic loci[18]. In support of this model, MM-BIR and BIR are prone to

[1]Division of Cancer and Genetics, School of Medicine, Cardiff University, Cardiff, UK. [2]Center for Chromosome Stability, Department of Cellular and Molecular Medicine, University of Copenhagen, Copenhagen, Denmark. [3]Department of Medicine, University of Virginia, Charlottesville, VA, USA. ✉e-mail: NgoG@cardiff.ac.uk; bairddm@cardiff.ac.uk

template switching in budding yeast studies[19,20]. However, the mechanism underlying chromoanasynthesis in human cells remains poorly understood due to technical challenges involved in detecting these events.

It is well established that telomere dysfunction contributes to chromothripsis via telomere fusion-driven chromatin bridge formation and DNA fragmentation[4,9,13,21]. Templated insertions have also been detected at these sites, suggesting the possible occurrence of chromoanasynthesis[13,15,21]. In support of this, CCRs can arise during telomere crisis in human cells in the absence of non-homologous end-joining (NHEJ) and these exhibit features that are consistent with a requirement for DNA replication during their formation[22]. However, the detection of CCRs is severely limited by the standard use of short-read DNA sequencing. To overcome this problem, we have developed a single-molecule, long-read DNA sequencing protocol and bioinformatic pipelines to detect and characterise CCRs originating from dysfunctional telomeres. Our approach reveals ultra-complex CCRs, resembling chromoanasynthesis, at fused telomeres and uncovers a highly mutagenic mitotic MM-BIR pathway as the driver of this enigmatic form of CCRs.

## Results

### Fusion-seq long-read (FSLR) uncovers CCRs in telomere crisis

We have previously developed single molecule PCR-based approaches to characterise de novo telomere fusion events in human cells[23]. DNA sequence characterisation revealed additional genomic sequences inserted between the fused telomeres[24]; however, a full understanding of the complexity of these events was limited by the technical constraints of short-read DNA sequencing. Thus, to fully characterise the genomic changes associated with telomere fusion, we established a modified version of the long-read amplicon sequencing protocol from Oxford Nanopore Technologies (ONT) with accompanying bioinformatic pipelines. With this new methodology, we were able to specifically sequence telomere fusion PCR products isolated from four primary human fibroblast cell lines (WI-38, MRC5, HCA2 and IMR90) undergoing a telomere-driven replicative crisis after 16.89 (WI-38), 29.27 (MRC5), 29.22 (HCA2) and 35.75 (IMR90) of population doubling from the point of senescence (Fig. 1a, b, Supplementary Fig. 1a, b, c, d).

We amplified telomere fusions from these fibroblast cell lines using a single-molecule PCR based assay that targets the unique 17p and XpYp telomeres, together with a family of telomeres that share sub-telomeric sequence similarity to the 21q telomere (Fig. 1a)[24]. To characterise these telomeric mutational events, we developed Fusion-Seq Long-Read (FSLR), a bioinformatic pipeline that allowed us to map and count each individual DNA segment identified within a fusion product with high accuracy (Supplementary Fig. 1e, f). A large majority (90–92%) of the identified telomere fusions display two alignments consistent with simple telomere fusion with no insertions (Fig. 1c, Supplementary Fig. 1g). Interestingly, however we also detected fusion events involving one or more insertions (i.e. alignment number >2, Fig. 1c, Supplementary Fig. 1g). For each fusion event the flanking DNA of the sequenced molecules mostly originated from the telomeres of 17p, XpYp and the 21q family of chromosomes targeted with the assay (Fig. 1d). The size range of the flanking sequences (Fig. 1e) were, as expected for 17p and XpYp telomeres, based on the positions of PCR primers used in the assay (Fig. 1a) relative to the start of telomere repeat array. Alignments longer than the sub-telomeric sequence size are indicative of a fusion within the remaining telomere repeats, while those shorter than the sub-telomeric sequence size define a deletion prior to fusion. These profiles are consistent with the profile of telomere fusion events described previously[23]. Interestingly, 21q family telomeres underwent large deletion before fusion. In contrast to the flanking DNA, inserted DNA originated from many genomic locations from all chromosomes (Fig. 1f, Supplementary Fig. 1h) and displayed a size range of 13 bp–18,065 bp (median = 167 bp, 234 bp,

155 bp and 208 bp for WI-38, MRC5, HCA2 and IMR90, respectively, Fig. 1g). Junctional analyses showed that 50.9% contained overlapping sequence (median = 2 bp), 11.4% were blunt and 37.7% contained insertions or gaps (sequence that could not be mapped accurately) (Supplementary Fig. 2a).

Next, we focused on the analysis of CCR events that we defined as those with greater than two insertions between sub-telomeres on a given fusion. The most complex fusion detected (complex 18) was 13,919 bp and contained 16 insertions from 9 different chromosomes, which were sandwiched between two chromosome 16q telomeres (Fig. 1h). The sizes of the insertions were between 55 and 3916 bp, and 12 of 17 junctions contained overlapping sequence (median = 3 bp, Supplementary Fig. 2b, Supplementary Data 1) consistent with the utilisation of a microhomology-mediated mechanism. The insertions in complex 18 originated from 14 different genomic loci of the genome (Supplementary Data 1), with the exception of two insertions (8i and 8ii) that originated from the same sequence on chromosome 8, albeit in opposite orientations (Fig. 1h, i). This was also true for the two flanking sub-telomeric DNA sequences (16i and 16ii) from chromosome 16 (Fig. 1h, i). We propose that the end of this molecule was generated from intra-strand fold-back-induced DNA synthesis, driven by annealing of an inverted repeat sequence (~700 bp) located on chromosomes 19 and 5 (Supplementary Fig. 2c).

Within our dataset of CCR events, we identified many foldback-induced DNA synthesis events: another example was observed within complex 12 that had a foldback at chromosome 6 (Fig. 1j, Supplementary Data 2), with insertion 6i and flanking DNA 17i copied from 6 v and 17ii, respectively (Fig. 1k). This event also contained four other insertions that originated from two insertion hotspots on chromosome 5 and 6 (Fig. 1j, k). Interestingly, the two insertions on chromosome 5 (5i and 5ii) occurred sequentially at the boundary of an inverted repeat sequence but with opposite orientation (Fig. 1k, Supplementary Fig. 2d). We propose that these insertions were generated from DNA replication of this locus, followed by template switching at the inverted repeat-induced cruciform structure (Supplementary Fig. 2d). The insertion hotspot on chromosome 6 (Fig. 1j, k) contains two insertions (6ii and 6iii) that are non-overlapping (Fig. 1k). These non-overlapping (or minimally overlapping) hotspots are frequently observed in complex insertion events and are often interspersed with insertions from other loci, as shown by another example, complex 13, which contains three such hotspots (Supplementary Fig. 2e, f, Supplementary Data 3). The presence of such hotspots suggests that these insertions were generated by non-random events occurring at these loci.

Overall, our novel nanopore telomere fusion sequencing technique facilitates the detection and quantification of CCRs at dysfunctional telomeres. The presence of microhomologies, foldback-induced DNA synthesis and template switching at inverted repeats suggests that these CCRs were generated through a process that is dependent on both microhomologies and DNA replication.

### Sub-telomeric DNA double-strand breaks induce CCRs resembling chromoanasynthesis

To investigate the underlying mechanisms of telomere dysfunction-induced CCRs, we employed Transcription Activator-Like Effector Nucleases (TALENs) to generate DNA double-strand breaks (DSBs) about 1.3 kb from the telomeres of the 21q and 16p families of related telomeres (Fig. 2a), between them encompassing at least 19 chromosome ends[25,26]. Consistent with our previous findings[26,27], we observed high frequencies of sub-telomeric DSBs and telomere fusions (Supplementary Fig. 2g, h) 1 day after transfection of TALEN-expressing plasmids into HCT116 wild-type (WT) cells. The length distributions of telomere fusion events, amplified at the single-molecule level using the single 21q1 PCR primer, revealed the presence of both deletions and numerous insertions (Fig. 2a, b).

 

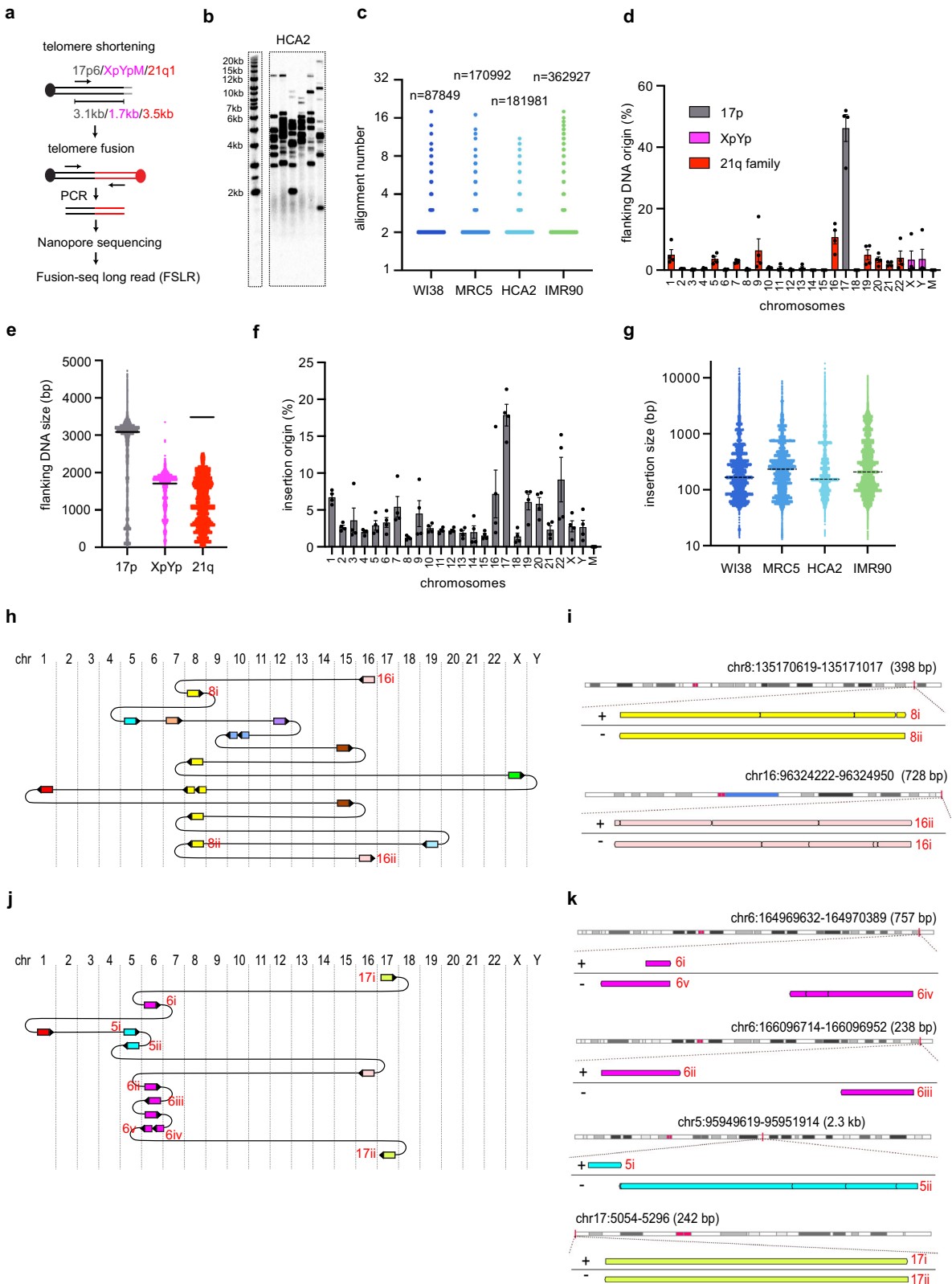

FSLR analysis revealed that the majority of TALEN-induced telomere fusions contained no insertions or were simple events, containing just one or two insertions (Fig. 2c). However, consistent with the analysis of fibroblasts undergoing telomere crisis, we could detect numerous complex events with three or more insertions, including rare ultra-complex events with more than twenty insertions (Fig. 2c). The level of complex insertions was much higher in HCT116 compared to fibroblasts undergoing telomere crisis or alternative lengthening of telomere positive U2OS cells transfected with 21q/16p TALENs (Fig. 2d). As expected, the flanking DNA originated from the 21q family of telomeres (Fig. 2e) and consisted predominantly of two sizes of ~1.9 kb or ~2.2 kb (Fig. 2f, flank), which represent DNA ends from the two major variants of 21q family that are unprocessed/minimally processed (Fig. 2a). Insertions had a size range of 12–16,854 bp

**Fig. 1 | Fusion-seq long-read (FSLR) uncovers CCRs in telomere crisis. a** A diagram of the telomere shortening-induced fusion assay, showing the position of primers and their distance to the start of telomere repeats (grey). **b** Telomere fusions were amplified using 17p6, XpYpM and 21q1 PCR primers from HCA2 fibroblasts transiting telomere crisis and visualised by Southern blot. This experiment was repeated independently with similar results for four times. **c** Scatter plot showing FSLR quantification of the number of alignments in the indicated number of telomere fusion molecules, which were isolated from the four fibroblasts indicated (alignment numbers are represented on a log2 scale, value of 2 represents fusion with no insertion, value ≥ 3 represent fusion with ≥ 1 insertion). The width of data point distribution is proportionate to the number of points at that Y value. **d** Quantification of the chromosomal origin of flanking DNA in telomere fusion molecules. Data plotted are means +/- standard error of the mean (SEM, n = 4 biological replicates), M mitochondrial DNA. **e** Scatter plot showing the size of

flanking DNA which originates from 17p (n = 201946), XpYp (n = 84857) and 21q family telomeres (n = 78009) from four biological replicates. The black lines indicate the expected size of the flanking DNA from PCR primer site up to the start of telomere repeats. **f** Quantification of the chromosomal origin of individual DNA insertions in telomere fusion molecules. Data plotted are means +/- standard error of the mean (SEM, n = 4 biological replicates). **g** Scatter plot showing the size of individual DNA insertions from WI38 (n = 11416), MRC5 (n = 16149), HCA2 (n = 21791) and IMR90 (n = 49017) with the median indicated by the dotted line. Diagrams showing how DNA molecules from different chromosomes are connected in complex 18 (**h**) and complex 12 (**j**). Direction of the arrows indicates the orientation of the DNA (right = +, left = -), chr chromosome. Genome browser (GW) plots showing the location of individual DNA alignments from complex 18 (**i**) or complex 12 (**k**) (highlighted with red text in Fig. 1h, j) at their mapped genomic loci. Source data are provided as a Source data file.

(median = 1566 bp, Fig. 2f, ins), 69% of which were derived from TALEN-expressing plasmids (Fig. 2g), while the remaining originated from all other chromosomal locations (Fig. 2g, Supplementary Fig. 2i). Junctional analyses showed that 56.1% of the junctions contained overlapping sequences (median = 3 bp), 15.8% were blunt and 28.1% contained insertions or unmapped gaps.

The most complex molecule detected, complex 111 (11,947 bp), has a remarkable 109 insertions from 18 different chromosomes, sandwiched between the two telomeres of chromosome 10 (Fig. 2h). The sizes of insertions were between 21 and 245 bp (median = 77 bp, Supplementary Data 4), and 51.8% of junctions contained microhomologies. 64 insertions originated from 19 hotspots (10 kb) which contained more than one insertion (Supplementary Data 5), and most of these contained non-overlapping insertions as described in Fig. 1k. However, 4 hotspots contained overlapping insertions, and importantly, 2 of these contained more than 3 overlapping insertions in a single event (Fig. 2h, i, j, Supplementary Data 5). These overlapping insertions can be best explained by a replicative origin, whereby the loci were replicated multiple times. These types of hotspots, with up to 10 overlapping insertions from a single locus, were frequently observed in complex molecules (Supplementary Fig. 3a, b, c, d, Supplementary Data 6, 7). Interestingly, some of these hotspots are flanked by DNA sequences such as inverted repeats or simple repeats that are predicted to form secondary structures (Fig. 2i, j, Supplementary Fig. 3b, d).

Two of the insertions from chromosome 9 in complex 111 (9i and 9ii) showed similar inverted-repeat-induced template switching events as described in Supplementary Fig. 2d, further supporting a replicative origin for these events (Fig. 2h, k). We also detected complex molecules exhibiting foldback-induced DNA synthesis events as described in Supplementary Fig. 2c. The most striking example, complex 6 with 4 insertions, involves a very long insertion of 16,854 bp from chromosome 22, which appears to have folded back on itself using 286 bp of inverted repeat sequence to initiate DNA synthesis towards the beginning of the molecule (Supplementary Fig. 3e, f, Supplementary Data 8).

Taken together, our long-read sequence analysis of CCR events obtained in cells undergoing a telomere-dependent replicative crisis or with TALEN-induced telomere fusions reveals several notable features, akin to those of chromoanasynthesis. These include a high level of junction microhomologies, overlapping insertions from hotspots, and frequent template switching events, which could be promoted by an error-prone replicative pathway similar to MM-BIR.

## Telomere dysfunction-induced chromoanasynthesis arises independently of LIG4, but requires LIG1/LIG3 and RAD52

Based on the data above, we hypothesise that an error-prone MM-BIR pathway, instead of end-joining, is responsible for generating chromoanasynthesis at dysfunctional telomeres. To examine this hypothesis, we tested the effect of deleting *LIG3*, *LIG4* or *RAD52*, which

promote microhomology-mediated-end-joining (MMEJ), non-homologous end-joining (NHEJ) or single-strand annealing (SSA) pathways, respectively. Consistent with previous findings, a *LIG4* knockout (KO) had a larger impact in reducing insertions compared to *LIG3* KO (Fig. 3a, Supplementary Fig. 4a)[26]. *LIG3:LIG4* double KOs behaved similarly to *LIG4* KO, whereas *RAD52* KO reduced insertions weakly compared to *LIG4* KO (Fig. 3a, Supplementary Fig. 4a).

Next, we examined how these KOs affected the complexity of insertion events using nanopore sequencing (Fig. 3b). *LIG3* KO did not affect fusion with simple (1 or 2) insertions but mildly reduced those with complex (3 or more) insertions. *LIG4* KO had the opposite effect by reducing simple insertions but not complex insertions. *LIG3:LIG4* double KOs did not further reduce complex insertion (compare to *LIG3* single KO) or simple insertion (compared to *LIG4* single KO). Surprisingly, *RAD52* KO strongly reduced the frequency of complex molecules, in addition to reducing simple fusions. Thus, the formation of simple insertions appears to be more dependent on LIG4, whereas complex insertion requires LIG3 and RAD52.

Next, we examined how the KOs affected the origin of insertions. A large proportion (67%) of simple insertions in WT cells originated from external DNA (comprising mostly TALEN-expressing plasmids, with small amount of copurified *E. coli* DNA), with the remaining 33% originating from endogenous chromosomes (Supplementary Fig. 4b). However, external DNA contributed to only 26% of all insertions in the complex events from WT cells (Supplementary Fig. 4b). *LIG4* KO (and *LIG3:LIG4* double KOs) significantly reduced the level of insertions that originated from external DNA, whereas *LIG3* KO or *RAD52* KO did not (Supplementary Fig. 4b, Supplementary Fig. 4c). This explains why *LIG4* KO and *LIG3:LIG4* double KOs significantly reduced the level of simple insertional events (Fig. 3b), as well as the size of simple insertions in these events (Fig. 3c), as external DNA insertions are longer than chromosomal DNA insertions (Supplementary Fig. 4d). *LIG4* KO (and *LIG3:LIG4* double KOs) also strongly reduced the number of blunt junctions and the length of sub-telomeric flanking DNA (Fig. 3d, e), likely due to increased resection in the absence of NHEJ[28]. In contrast, *LIG3* KO and *RAD52* KOs slightly increased blunt junctions and flanking DNA size, possibly due to increased usage of NHEJ (Fig. 3d, e). We propose that a large proportion of simple insertions involve ligation of DSB ends with linearised TALEN plasmids by NHEJ, whereas formation of more complex insertions requires another pathway that is less dependent on LIG4. In support of this hypothesis, we identified many CCR events in *LIG4* KO cells that showed the hallmarks of MM-BIR, including microhomologies and insertion hotspots with non-overlapping or overlapping insertions (Fig. 3f, g, Supplementary Data 9).

LIG1 acts redundantly with LIG3 in MMEJ, and also in BIR, where they are required for ligating DNA during second strand synthesis[29,30]. To examine the effect of inhibiting both ligases, we treated *LIG3* KOs with a LIG1 inhibitor, L82-G17[31]. Consistent with the result of *LIG3* KO,

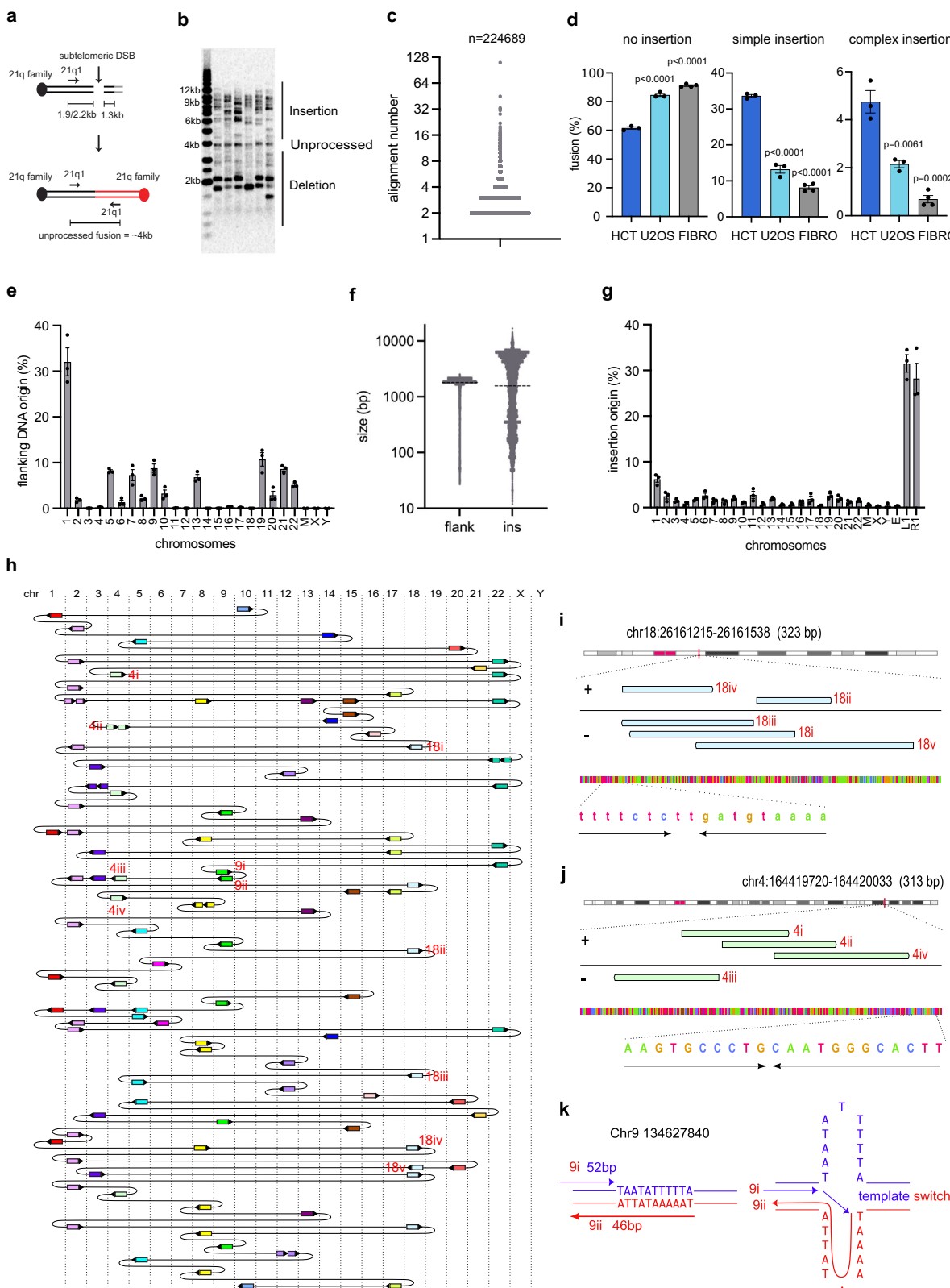

L82-G17 specifically reduced complex insertions while having no effect on simple insertions (Fig. 3h, i). Interestingly, L82-G17 increased the size of both simple and complex insertions but did not affect junction type or external DNA level (Fig. 3j, Supplementary Fig. 4e, f). We conclude that simple insertions at dysfunctional telomeres depends on LIG4-mediated NHEJ, whereas chromoanasynthesis depends on another pathway which require RAD52 and LIG1/LIG3.

## Chromoanasynthesis is promoted by Polθ and suppressed by the DNA damage checkpoint

To identify a DNA polymerase responsible for telomere dysfunction-induced chromoanasynthesis, we examined the involvement of Polθ, which initiates DNA synthesis at DNA junctions with microhomology in the same pathway as LIG1/3, and is prone to template switching[32,33]. Importantly, a RPE1 *POLQ* KO accumulated fewer insertions at TALEN-

**Fig. 2 | Sub-telomeric DNA double-strand breaks (DSBs) induce CCR with a replicative origin. a** A diagram of the TALEN-induced telomere fusion assay showing the position of primers and the expected amplicon size. **b** Telomere fusion assay using DNA isolated from HCT116 cells 2 days after TALEN nucleofection (repeated independently with similar results for three times). **c** Scatter plot showing FSLR analysis of telomere fusion. The width of data point distribution is proportionate to the number of points at that Y value. **d** Bar chart comparing telomere fusion with no insertion, simple (1 or 2) insertion or complex (3 or more) insertion in HCT116, U2OS and fibroblasts (FIBRO). Data plotted are means +/- s.e.m (n = 3 biological replicates for HCT116 and U2OS, n = 4 biological replicates for fibroblasts). *P* values were obtained using Student's t-test (unpaired two-tailed, equal variance). The fibroblast data were the same as in Supplementary Fig. 1g. **e** Quantification of the chromosomal origin of flanking DNA. M mitochondrial DNA. Data plotted are means +/- s.e.m (n = 3 biological replicates). **f** Scatter plot showing the size of flanking DNA (flank, n = 347237) and insertions (ins, n = 152849), dotted line = median. **g** Quantification of the origin of individual DNA insertions (L1 and R1 = TALEN left and TALEN right plasmid, E = *E.coli* DNA). Data plotted are means +/- s.e.m (n = 3 biological replicates). **h** Diagram showing how DNA molecules from different chromosomes are connected in complex 111. Direction of the arrows indicates the orientation of the DNA (right = +, left = -), chr chromosome. **i, j** Genome browser (GW) plot showing the location of each individual DNA alignment from complex 111 (highlighted with red text in Fig. 2h) at their mapped genomic loci. The four-colour bar at the bottom of each panel represents the DNA sequences at the locus (red = T, blue = C, orange = G, green = A). Flanking DNA sequence predicted to form a secondary structure are indicated by the lines with arrowheads. **k** Diagram showing two insertions (highlighted with red text in Fig. 2h), which may have been generated by a template switching mechanism at an inverted repeat sequence. Source data are provided as a Source data file.

induced DSBs compared to WT (Fig. 4a, Supplementary Fig. 5a), and the proportion of complex insertions (in addition to simple insertions) observed in the *POLQ* KO was also significantly reduced (Fig. 4b). This was not due to differences in the cell cycle (Supplementary Fig. 5b). Interestingly, loss of Polθ had little impact on the level of junction overlaps but reduced the size of microhomologies (Supplementary Fig. 5c, d). The *POLQ* KO slightly reduced the size of simple insertions, possibly due to reduced TALEN insertions (Fig. 4c, Supplementary Fig. 5e). Intriguingly, however, while the number of complex insertions in a *POLQ* KO decreased, the size of the complex insertions that did occur increased (Fig. 4c). We conclude that Polθ plays an important role in this replicative pathway.

Polθ facilitates the repair of DSBs in mitosis following activation by PLK1-induced phosphorylation and CDK1/PLK1-induced phosphorylation of RHINO[34–36]. To confirm whether telomere dysfunction-induced CCRs occur in mitosis, we examined the effect of forcing premature cellular entry into mitosis in the presence of DSBs, which would be expected to increase the number of DSBs being repaired by activated Polθ in mitosis. To do this, we inactivated the G2/M DNA damage checkpoint pathway regulated by ATR, CHK1 and WEE1[37]. WEE1 and CHK1 inhibitors caused an increase of cell accumulation in G1 2 days following the induction of DSBs, which was accompanied by a decrease of cells in G2/M (Supplementary Fig. 5f). This is likely due to more cells entering and eventually exiting mitosis before becoming arrested in the next G1. The ATR inhibitor had a minimal effect on cell cycle progression, possibly due to a compensatory function of ATM. All three inhibitors increased the overall frequencies of telomere fusion (Fig. 4d), but importantly only WEE1 and CHK1 inhibitors increased the frequencies of complex insertions (Fig. 4e), supporting the hypothesis that these MM-BIR-mediated CCRs occur in mitosis. To examine the role of ATM, we treated cells with an ATM inhibitor, KU-60019 and observed increased numbers of cells in G1, accompanied by a reduction of cells in G2/M, 1 day following the induction of DSBs (Supplementary Fig. 6a). This was similar to cells treated with WEE1 or CHK1 inhibitors and suggest that more cells progressed through mitosis due to the absence of G2/M checkpoint. However, cells treated with ATM inhibitor continued to progress through the cell cycle, likely due to the absence of G1 checkpoint, and by day 2, there were more cells in G2/M compared to DMSO treated cells (Supplementary Fig. 6a). ATM inhibition strongly reduced fusions associated with deletion and increased blunt junctions (Supplementary Fig. 6b, c, d), likely due to its role in promoting DNA resection[38]. Importantly, ATM inhibition increased the frequency of complex insertions, further supporting the role of G2/M checkpoint machinery in suppressing chromoanasynthesis (Supplementary Fig. 6e).

Strikingly, we identified an ultra-complex fusion event in cells treated with the WEE1 inhibitor (Fig. 4f, g, Supplementary Data 10). Complex 134 (40,146 bp) had 132 insertions, with 127 of them originating from a single locus on chromosome 17. This locus contains three inverted repeats that have the potential to form cruciform structures and induce inter-strand template switching, as described earlier (Figs. 1k and 2k, Supplementary Figs. 2d, 7a). Polθ goes through cycles of DNA annealing, synthesis and dissociation[33]. We propose that these template switching events in complex 134, complex 111 (Fig. 2h) and complex 12 (Fig. 1j) were generated by Polθ, which upon entering the cruciform structure, stalled and switched template by promoting the annealing of synthesised single-stranded DNA (ssDNA) to the opposite side of the cruciform, before initiating DNA synthesis of the opposite strand until it stalled and switched templates again at the next cruciform (Supplementary Fig. 7b). We identified three rounds of this 'cruciform trapped' DNA replication in this locus in complex 134, generating 93, 14 and 20 insertions, respectively, which were interrupted by the DNA briefly leaving the locus twice. Thus, this MM-BIR pathway led to a striking >100x amplification of a locus in one single event. Interestingly, this locus encodes TMEM105, a putative oncogene[39]. Complex 134 has two identical flanking sequences from chromosome 7 joined to two identical sequences from chromosome 13 (Fig. 4f, Supplementary Data 10), indicative of a foldback event also known to be catalysed by Polθ[40,41], which supports a replicative origin of this event as well.

Intriguingly, in addition to increasing MM-BIR events, we detected in cells treated with the WEE1 inhibitor, insertions that exhibited a unique pattern where multiple insertions line up perfectly with minimal gaps (8iii, 8iv and 8v in Supplementary Fig. 7c, d, Supplementary Data 11, 17ii, 17iii, 17iv and 17v in Supplementary Fig. 7e, f, Supplementary Data 12). This pattern can be best explained by a 'fragmentation and ligation' model rather than a replicative model, suggesting that pushing more cells into mitosis prematurely in the presence of DSBs may increase the chance of localised DNA fragmentation, or that WEE1 actively inhibits nuclease activity involved in this fragmentation process, as described[42].

To examine whether the increased frequency of telomere dysfunction-induced CCR events in cells subjected to WEE1 inhibition were driven by Polθ, we treated cells with WEE1 inhibitors together with ART558 or novobiocin, which inhibit the DNA polymerase and helicase activity of Polθ, respectively[43,44]. Both inhibitors strongly reduced the level of complex insertions, indicating that Polθ stimulates MM-BIR in mitosis using both its helicase and polymerase activities (Supplementary Fig. 8a). These inhibitors mildly affected junction type but reduced the microhomology size (Supplementary Fig. 8b, c), as observed in *POLQ* KO cells. Polθ inhibition also increased the insertion size in complex events as observed in *POLQ* KO cells (Fig. 4c, Supplementary Fig. 8d), suggesting the possible involvement of a more processive DNA polymerase.

Next, we examined whether Polθ inhibition also reduced complex insertions in the absence of RAD52 (Fig. 4h). Importantly, ART558 further reduced complex insertion in *RAD52* KO cells, showing that Polθ promotes chromoanasynthesis independently from RAD52

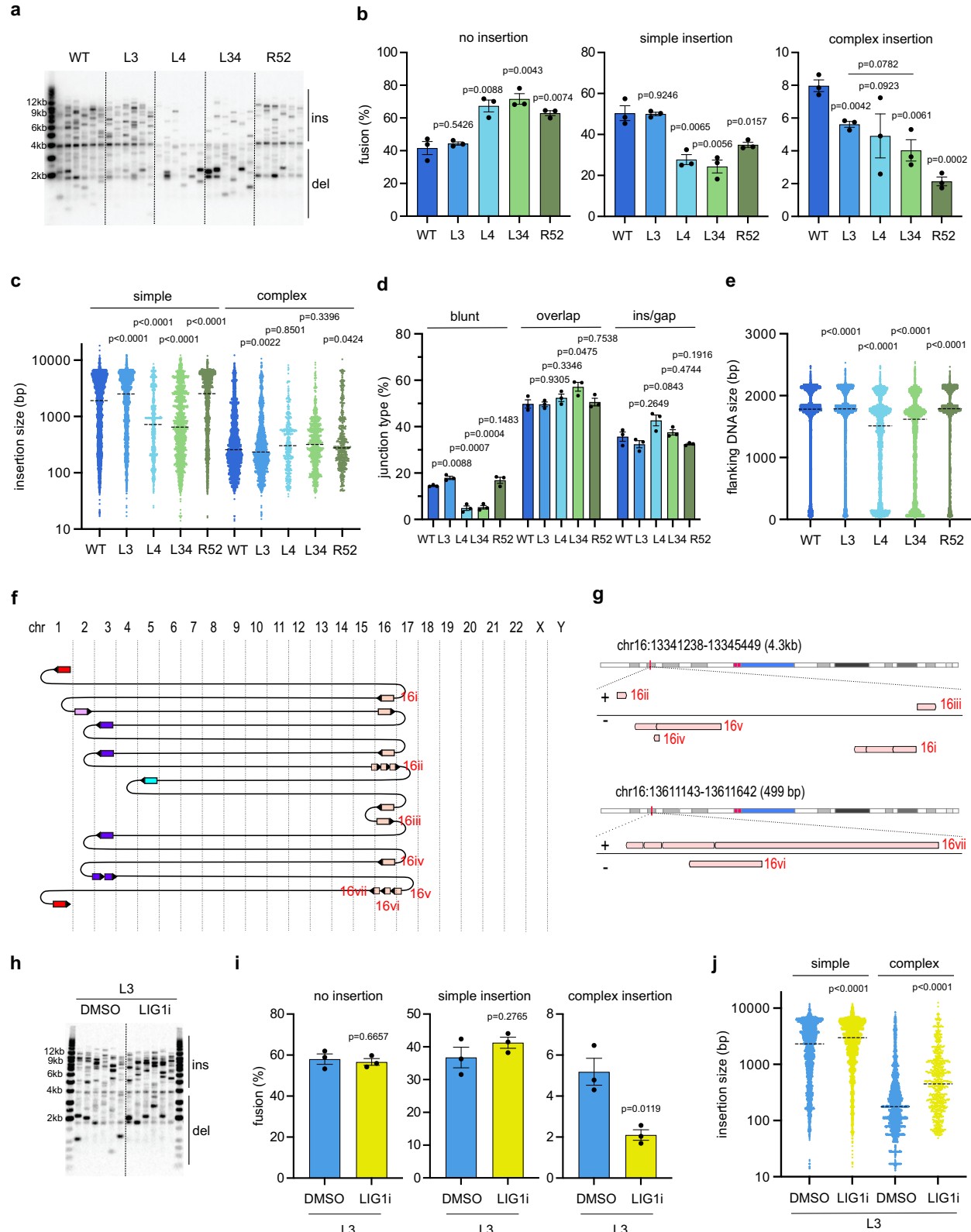

(Fig. 4i). ART558 reduced MH size and increased blunt junctions as observed in WT cells (Fig. 4j, Supplementary Fig. 8e) but intriguingly did not affect insertion size of complex events (Supplementary Fig. 8f). Taken together, we conclude that Polθ contributes to chromoanasynthesis in a RAD52-independent pathway. This highly mutagenic pathway could occur in mitosis but is usually suppressed by the G2/M DNA damage checkpoint.

## Polθ and Polδ initiate MM-BIR in mitosis

Polθ facilitates MH-mediated pairing and DNA synthesis at DNA ends, but it has limited processivity[33,45] and thus is unlikely to be responsible for the extensive DNA replication required for the generation of the long (>10 kb) insertions observed in some of the CCRs observed here. However, Polθ can recruit Polδ, a more processive DNA polymerase, to extend DNA synthesis initiated by Polθ and facilitate

**Fig. 3 | CCRs arise independently of LIG4 but requires LIG1/3 and RAD52.**
**a** Telomere fusion blot showing fusion amplicons from HCT116 wild-type (WT),
*LIG3* KO (L3), *LIG4* KO (L4), *LIG3:LIG4* Double KO (L34) and *RAD52* KO (R52) cells (ins
= insertion, del = deletion). This experiment was repeated independently with
similar results for three times. **b** Bar chart comparing telomere fusion with no
insertion, simple (1 or 2) insertion or complex (3 or more) insertion. Data plotted
are means +/- s.e.m (n = 3 biological replicates). *P* values were obtained using Stu-
dent's t-test (unpaired two-tailed, equal variance). **c** Scatter plot showing the size of
individual DNA insertions from three biological replicates (n = 6331, 4165, 969,
1788, 2580, 3400, 1785, 419, 1039, 559 from left to right, dotted line = median).
*P* values were obtained using two-tailed Mann-Whitney test. **d** Quantification of
junction type. Data plotted are means +/- s.e.m (n = 3 biological replicates). *P* values
were obtained as described in Fig. 3b. **e** Scatter plot showing the size of flanking
DNA from three biological replicates (n = 19623, 13370, 5535, 12740, 11485 from left

to right, dotted line = median). *P* values were obtained as described in Fig. 3c.
**f** Diagram showing how DNA molecules from different chromosomes are con-
nected in complex 21, isolated from HCT116 *LIG4* KO. Direction of the arrows
indicate the orientation of the DNA (right = +, left = -), chr = chromosome.
**g** Genome browser (GW) plot showing the location of each individual DNA align-
ment (highlighted with red text in Fig. 3f) at their mapped genomic loci. **h** Telomere
fusion blot showing telomere fusion amplicons from HCT116 *LIG3* KO (L3) treated
with DMSO or LIG1 inhibitor (LIG1i). This experiment was repeated independently
with similar results for three times. **i** Bar chart comparing telomere fusion com-
plexity. Data plotted are means +/- s.e.m (n = 3 biological replicates). *P* values were
obtained as described in Fig. 3b. **j** Scatter plot showing the size of individual DNA
insertions from three biological replicates (n = 4206, 3699, 2691, 740 from left to
right, dotted line = median). *P* values were obtained using two-tailed Mann-Whitney
test. Source data are provided as a Source data file.

---

MMEJ, which require gap filling of >70nt[46]. How Polδ activity is
restricted to short DNA synthesis before ligation in MMEJ remains
unclear. As Polδ is essential for BIR[30,47–49], we hypothesise that
recruitment of Polδ by Polθ to MH-paired ends may initiate extensive
DNA synthesis similar to BIR, i.e. MM-BIR. To test this hypothesis, we
examined whether Polθ can stimulate Mitotic DNA Synthesis
(MiDAS), a Polδ-dependent BIR pathway initiated at collapsed repli-
cation forks in mitosis[50].

To do this, we treated U2OS WT cells, which are known to utilise
MiDAS to repair collapsed replication forks at telomeres[51], with a low
dose of aphidicolin in S phase and monitored EdU incorporation in
early mitosis by following an established protocol[52–54]. Importantly,
treatment of cells with a Polθ inhibitor, ART558, reduced MiDAS
(Fig. 5a, b, c), confirming that Polθ can initiate BIR at a subset of col-
lapsed replication forks in mitosis. To determine whether Polθ acts
specifically to promote MM-BIR at telomeres, we performed MiDAS
coupled with telomere-FISH (Fluorescent in situ hybridisation) analysis
(Fig.5d, e). Polθ inhibition reduced telomeric MiDAS (Fig. 5f), in addi-
tion to non-telomeric MiDAS (Supplementary Fig. 9a), which shows
that Polθ acts on both telomeric and non-telomeric loci to initiate BIR
in mitosis. Interestingly, RAD52 has also been implicated in initiating
MiDAS[52], supporting our hypothesis that Polθ and RAD52 stimulates
MM-BIR and chromoanasynthesis in mitosis.

To identify the processive DNA polymerase responsible for this
MM-BIR pathway, we next examine the role of Polδ by depleting
POLD1, which is the catalytic subunit of Polδ. In parallel, we also
depleted Polθ and POLE1, the catalytic subunit of Polε, another pro-
cessive DNA polymerase involved in DNA replication. Depletion of all
three polymerases slightly reduced the frequencies of telomere fusion
(Fig. 5g, Supplementary Fig. 9b, c). Consistent with results obtained
with both *POLQ* KO and inhibitors, depletion of Polθ reduced the level
of complex insertions and microhomology size, as well as increasing
the size of insertion of complex events (Fig. 5h, i, Supplementary
Fig. 9d, e). Importantly, depletion of POLD1, but not POLE1, reduced
the level of complex molecules, suggesting that POLD1 supports Polθ
in this MM-BIR pathway (Fig. 5h). In further support of this conclusion,
POLD1 depletion reduced microhomology size and increased the fre-
quency of blunt junctions, as was observed in Polθ-depleted cells
(Supplementary Fig. 9e, f). Intriguingly, POLD1 depletion induced a
similar increase in insertion size in complex events as observed fol-
lowing Polθ depletion, whereas POLE1 depletion significantly reduced
the insertion size of complex events, which were not due to differences
in the level of external DNA (Fig. 5i, Supplementary Fig. 9g). We pro-
pose that while Polθ and Polδ are the dominant polymerases that
participate in this mitotic MM-BIR pathway, a fraction of the events
utilise Polε, which contributes to some extensive DNA replication
events and long insertions. We conclude that chromoanasynthesis is
generated by a mitotic MM-BIR pathway which is dependent on Polθ
and Polδ.

## Mitotic MM-BIR is regulated by processivity factors essential for BIR and represents a novel source of CCRs

To further confirm the involvement of MM-BIR in chromoanasynthesis,
we examined the requirement for PIF1 and POLD3, two processivity
factors specifically required for extensive DNA synthesis by Polδ dur-
ing BIR. PIF1 is essential for establishing migrating D-loop structures,
whereas POLD3 stabilises Polδ on DNA[47–49,55]. We employed CRISPR-
Cas9 to target *PIF1* and *POLD3* loci in RPE1-hTERT cells, but could only
recover *POLD3* heterozygote and *PIF1* mutant clones with residual PIF1
activity, suggesting that these genes are essential in this cellular
background. We employed siRNA to further deplete PIF1 and POLD3 in
these clones (Supplementary Fig. 10a, b) and observed slight enrich-
ment of cells in the G1 (siPIF1) or S phase (siPOLD3), accompanied by a
concomitant reduction of cells in G2/M, following the depletion of
these proteins (Supplementary Fig. 10c). PIF1 depletion reduced the
level of deletions and increased blunt junction, likely due to its role in
promoting DNA resection[56], whereas POLD3 depletion slightly reduced
the overall frequency of fusion (Fig. 6a, Supplementary Fig. 10d, e).
FSLR analysis showed that PIF1 depletion strongly increased the level
of fusion with simple insertion at the expense of fusion without
insertion, while having no effect on complex insertion (Fig. 6b).
Importantly however, insertion sizes of complex events were sig-
nificantly shorter following PIF1 depletion (Fig. 6c), which was not due
to a difference in external DNA insertion (Supplementary Fig. 10f),
supporting the hypothesis that long insertions in complex events are
generated by PIF1-initiated BIR. In contrast to PIF1, POLD3 depletion
increases the level of complex insertion events (Fig. 6b), supporting a
role for this protein in maintaining the stability of Polδ in MM-BIR,
which suppress frequent template switching. POLD3 depletion
increased the insertion size of complex fusion (Fig. 6c), similar to what
was observed in Polθ- and POLD1-depleted cells. Interestingly, POLD3
depletion did not increase complex insertion in the absence of POLQ
(Supplementary Fig. 10g, h, i), suggesting that frequent template
switching observed following POLD3 depletion is due to Polθ.

Next, we examine the requirement of PCNA, a processivity factor
for Polδ and Polε in DNA replication, which is essential for PIF1 activity
in BIR[55,57]. PCNA depletion by siRNA (Supplementary Fig. 11a) did not
cause a big difference in cell cycle distribution (Supplementary
Fig. 11b), telomere fusion spectrum or the level of complex insertions
(Fig. 6d, e). However, importantly, similar to what was observed in PIF1
depleted cells, PCNA depletion strongly reduced the insertion size in
complex events, (Fig. 6f), which was not due to a difference in external
DNA insertion (Supplementary Fig. 11c), supporting the hypothesis
that long insertions are driven by extensive DNA synthesis facilitated
by PCNA. We conclude that DNA synthesis and template switching in
chromoanasynthesis are regulated by POLD3, PIF1 and PCNA.

The two current known drivers of CCRs are the formation of
chromatin bridges and micronuclei following errors in chromosome
segregation, which can lead to chromosome fragmentation and

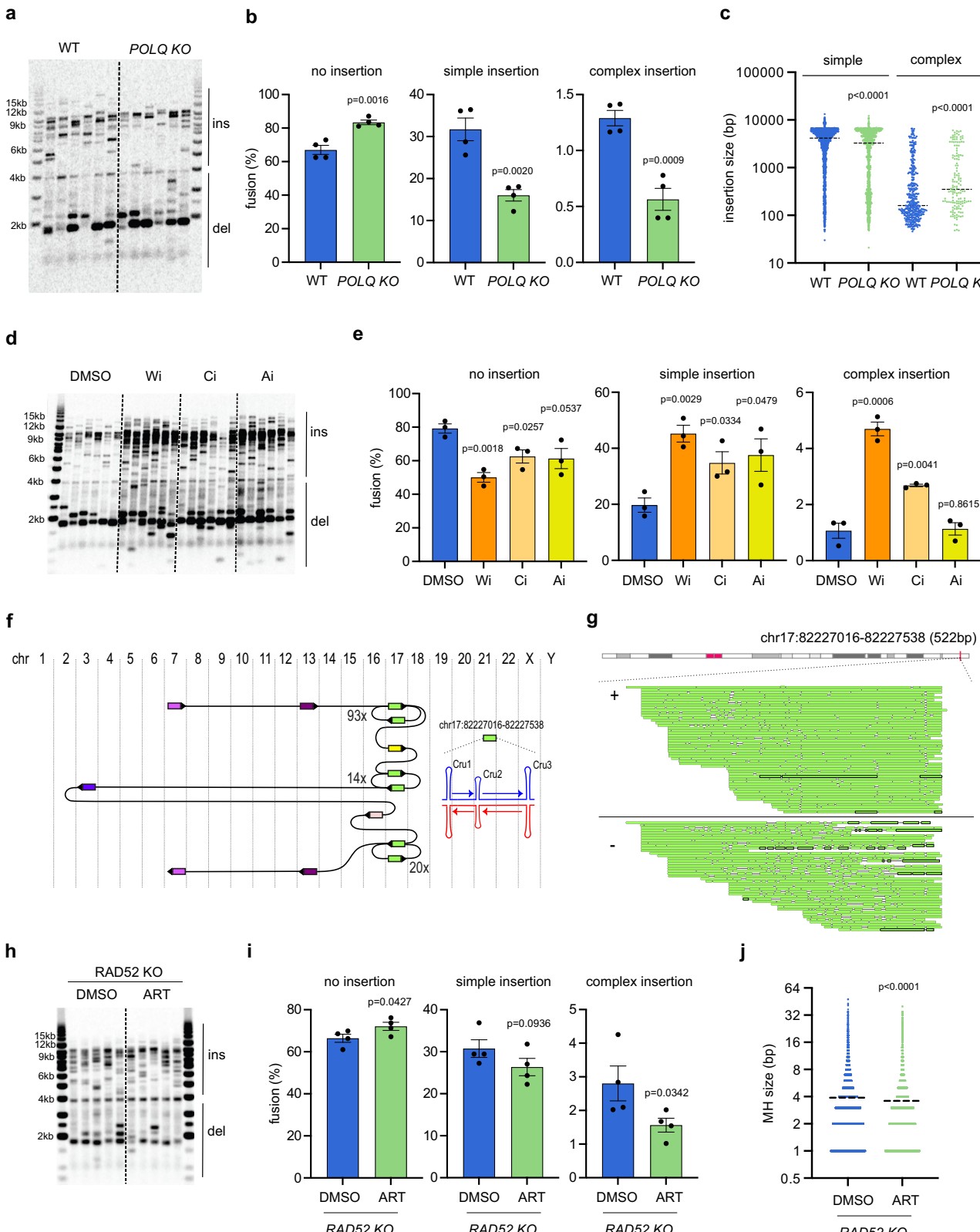

chromothripsis[1,58]. To examine whether mitotic MM-BIR acts independently of these events, we arrested cells in prometaphase using nocodazole in order to prevent entry into anaphase, where bridges and micronuclei form. Importantly, a failure to transit mitosis (Supplementary Fig. 11d) had no effect on the frequency of telomere fusion or complex insertions at these DSBs (Fig. 6g, h), and it did not affect insertion size, the level of external DNA or junction type (Fig.6i,

Supplementary Fig. 11e, f). We propose that MM-BIR occurs in early mitosis to generate telomere dysfunction-induced chromoanasynthesis independently of the formation of chromatin bridges or micronuclei.

## Discussion

Here, we established a single-molecule, long-read DNA sequencing protocol to reveal the structure of catastrophic CCRs at dysfunctional

**Fig. 4 | Chromoanasynthesis is promoted by Polθ and suppressed by the DNA damage checkpoint. a** Telomere fusion blot showing telomere fusion amplicons from RPE1 wild-type (WT) or *POLQ* KO (ins insertion, del deletion). This experiment was repeated independently with similar results for four times. **b** Bar chart comparing telomere fusion with no insertion, simple (1 or 2) insertion or complex (3 or more) insertion. Data plotted are means +/- s.e.m (n = 4 biological replicates). *P* values were obtained as described in Fig. 3b. **c** Scatter plot showing the size of individual DNA insertions from four biological replicates (n = 1979, 1218, 416, 142 from left to right, dotted line = median). *P* values were obtained using two-tailed Mann-Whitney test. **d** Telomere fusion blot showing telomere fusion amplicons from RPE1 WT cells treated with DMSO or a WEE1 inhibitor (Wi), a CHK1 inhibitor (Ci) or an ATR inhibitor (Ai) (ins insertion, del deletion). This experiment was repeated independently with similar results for three times **e** Bar chart comparing telomere fusion molecules. Data plotted are means +/- s.e.m (n = 3 biological replicates). *P* values were obtained using as described in Fig. 3b. **f** Diagram showing

how DNA molecules from different chromosomes are connected in complex 134. Direction of the arrows indicate the orientation of the DNA (right = +, left = -), chr = chromosome. All chromosome 17 alignments in green originated from a single locus, which contained three inverted repeat sequences capable of forming cruciform (Cru) structures as indicated. **g** Genome browser (GW) plot showing the genomic location of chromosome 17 alignments from complex 134. **h** Telomere fusion blot showing telomere fusion amplicons from HCT116 *RAD52* KO treated with DMSO or Polθ inhibitor (ART558). This experiment was repeated independently with similar results for four times. **i** Bar chart comparing telomere fusion molecules. Data plotted are means +/- s.e.m (n = 4 biological replicates). *P* values were obtained using Student's t-test (unpaired one tailed, equal variance). **j** Scatter plot comparing the size of microhomology (MH) at junctions from four biological replicates (n = 17663, 12383 from left to right, dotted line = means). *P* values were obtained using two-tailed Mann-Whitney test. Source data are provided as a Source data file.

telomeres. Remarkably, the CCRs observed share all the hallmarks of chromoanasynthesis, including microhomologies, templated insertions and frequent template-switching[10,11,13,15]. MM-BIR has long been invoked to explain the enigmatic process of chromoanasynthesis[10], but it has been technically challenging to directly investigate this pathway in human cells. Here, we showed that a mitotic MM-BIR pathway, which is dependent on RAD52, Polθ and Polδ, and tightly regulated by POLD3, PIF1 and PCNA, appears to be responsible for this highly mutagenic process.

In a mitotic MM-BIR model (Fig. 7), DNA ends utilise the strand annealing activities of RAD52 or Polθ[59–61] to bind to exposed DNA with microhomologies in mitosis (Fig. 7a) and initiate end bridging DNA synthesis by Polθ (Fig. 7b). The DNA synthesis initiated is short but becomes more processive following the recruitment of Polδ by Polθ or RAD52[46,52], and the arrival of processivity factors POLD3, PIF1 and PCNA (Fig. 7c). However, the MM-BIR pathway initiated is highly unstable, leading to frequent inter-chromosomal (Fig. 7d), inter-strand (Fig. 7e), or intra-strand/fold-back template switching (Fig. 7f), and this process appears to continue until the DNA is ligated with another DNA end or stabilised by telomere addition. LIG1/3 is required at the final telomere ligation step (Fig. 7g), and could also contribute to joining discontinuous DNA synthesised on the second strand by Polδ[30] (Fig. 7h), which is required for completion of MM-BIR.

BIR and MM-BIR utilise template switching[19,20,60,62], but we propose that mitotic MM-BIR can lead to higher levels of template switching, as observed in chromoanasynthesis, because of four factors: (1) the suppression of homologous recombination and NHEJ in mitosis, which limits DNA repair capacity[63], and allows for continuous cycles of template switching to occur, (2) high level of mitotic Polθ activity[34–36], which is prone to dissociation and template switching[32,33,45], and could potentially disrupt the processivity of Polδ in DNA replication due to frequent switching between the two polymerases at DNA ends, as observed in MMEJ[46], (3) templates may constitute difficult-to-replicate genomic loci prone to secondary structure formation and 4) highly condensed chromosomes are not conducive for processive DNA replication. POLD3 is particularly important in suppressing template switching in MM-BIR, likely by preventing dissociation of Polδ through increasing its affinity to PCNA[64].

Despite repeated template switching, mitotic MM-BIR frequently returns to the same template, as observed in chromoanasynthesis[13,15]. We propose that the newly replicated DNA may engage and copy from a new template while still linked with a previous template (Fig. 7d). This model is similar to the multi-invasion-induced rearrangement mechanism observed in yeast, where a DNA end simultaneously invades two donor templates[65]. It is also possible that there are limited unreplicated DNA templates that maintain an open and accessible chromatin structure in mitosis, or that mitotic chromosomal entanglement brings certain loci into close proximity[66]. We also discovered inverted repeat-induced trapping of MM-BIR on the same template,

which in extreme form, can lead to >100× amplification of a single genomic locus.

Despite the various hindrances to DNA replication during mitosis, we could detect replication of long stretches of DNA (>15 kb) during mitotic MM-BIR. Analysis of frequent DNA foldback events indicates that up to ~17 kb of ssDNA can be exposed during MM-BIR (Supplementary Fig. 3e), which implies that there could be a long delay between the first and second strand DNA synthesis, as proposed for the migrating bubble hypothesis for BIR[49,67]. The involvement of PIF1 and PCNA further supports the formation of migrating D-loops during MM-BIR[49,55,57]. While Polθ and Polδ are the dominant DNA polymerases which participate in mitotic MM-BIR, Polε may also contribute to this pathway, as supported by the reduced insertion size observed in the absence of this polymerase and the increased insertion size in the absence of Polθ or Polδ. We propose that Polε may contribute to MM-BIR following a Polδ to Polε switch mediated by PCNA[68]. Another intriguing possibility is that during mitotic MM-BIR, some D-loops get cleaved and converted into regular replication forks. The involvement of non-processive and processive DNA polymerases in MM-BIR could help explain the co-occurrence of short and long insertions as observed in chromoanasynthesis[8,14,15,17].

A larger question is why would cells ever utilise a mitotic MM-BIR pathway, which is extremely mutagenic? The answer appears to be that it is done *quasi* perforce as the Polθ pathway represents the failsafe mechanism to repair DSBs before cell division[32,34–36]. In support of this, we found that mitotic MM-BIR is strongly suppressed by the G2/M DNA damage checkpoint (Fig. 4e). Interestingly, telomeres become deprotected during mitosis[69], and this may possibly explain why shortened telomeres engage this repair pathway when experiencing telomere crisis (Fig. 1) or sub-telomeric DSBs (Figs. 2 and 5). Intriguingly, we showed that in addition to promoting mitotic MM-BIR at dysfunctional telomeres/DSBs, Polθ also stimulates mitotic DNA synthesis (MiDAS) at telomeric and non-telomeric loci in response to replication stress. The MiDAS model envisages broken replication forks initiating a BIR-like pathway to restore the fork and complete replication, but its impact on genome stability remains unclear[50]. The involvement of Polθ in MiDAS suggests that this pathway could potentially lead to chromoanasynthesis and would thus be highly mutagenic.

A recent study based on short-read sequencing showed that Polθ promotes the formation of MMEJ-induced deletion at common fragile sites after entry into mitosis, but this was independent of MiDAS[70]. Interestingly, some Polθ-dependent templated insertions were detected at the breakpoint junctions, but these were mostly simple and short (<20 bp). Further investigation is warranted to clarify the role of Polθ in promoting mitotic MM-BIR or MMEJ at different regions of the genome, and to understand how this process is regulated. Despite the involvement of mitotic MM-BIR in chromoanasynthesis, our model does not completely rule out a role for MMEJ-induced ligation. Due to the presence of MMEJ factors Polθ and Polδ, it is possible that once

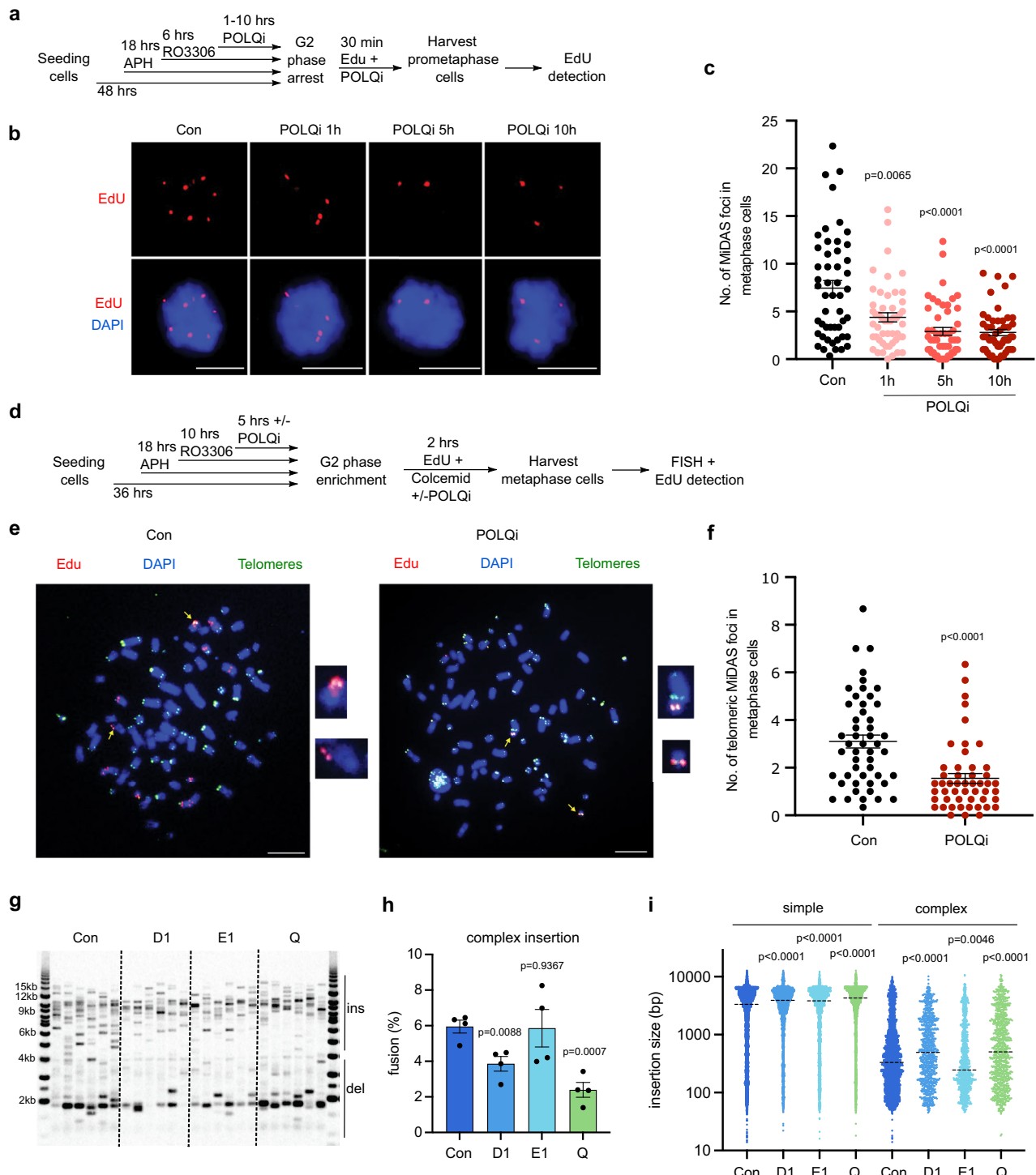

DNA ends engaged in MM-BIR dissociate from their template, they can switch to an MMEJ mode and ligate with other DNA fragments. In support of this hypothesis, unique insertion patterns observed in WEE1 inhibitor treated cells show that short, fragmented DNA can be ligated into some CCR events (Supplementary Fig. 7c–f). Furthermore, most of the MM-BIR events detected here rely on MMEJ to ligate with another telomere and complete the repair process. The tight association between MM-BIR and MMEJ proteins could explain the exclusive requirement of LIG1/3, instead of LIG4, in MM-BIR. This plasticity between MM-BIR and MMEJ usage could contribute to the co-occurrence of chromoanasynthesis and chromothripsis[8,13,14,17,21]. However, the detection of such events, especially megabase-scale DNA

fragmentation and ligation described in chromothripsis, is difficult in our assay due to the amplification limit of PCR.

Currently, micronuclei and chromatin bridges represent two known sources of CCR, especially chromothripsis[1,58]. We found that mitotic MM-BIR can generate CCRs in early mitosis independently of these two processes, thus revealing a new source of CCRs. However, we could not rule out the involvement of these two processes in further amplifying CCRs together with MM-BIR. Our results show that this MM-BIR pathway could specifically amplify the under-replicated region of the genome in mitosis implicated in MiDAS, which are actively transcribed genomic domains, most of which are genic regions experiencing replicative stress[71,72]. This pathway thus provides a

**Fig. 5 | Polθ and Polδ initiates MM-BIR in mitosis. a** Experimental workflow for analysis of MiDAS in prometaphase U2OS cells following aphidicolin (APH) treatment, enrichment of G2 cells with RO3306 and Polθ inhibition with ART558. Hr hour, min minute. Representative images (**b**) and quantification (**c**) of MiDAS foci (labelled with EdU; red) in prometaphase cells treated as shown in (**a**). DNA was stained with DAPI (blue). Scale bars, 10 μm. In each quantification, data are presented as mean +/- s.e.m of three independent experiments. In total, 150 cells were analyzed for each condition (n = 150). *P* values were calculated with a two-tailed non-parametric Mann–Whitney test. **d** Experimental workflow for analysing MiDAS in metaphase spreads of U2OS cells in parallel with telomere detection using a PNA-telomere probe. **e** Representative immunofluorescence images of MiDAS (labelled with EdU, red) and telomeres (green) in cells without (left) or with Polθ inhibition (right) at early M phase. Yellow arrows indicate the positions of the zoomed images where EdU is co-localised with telomere PNA-probe. Scale bars, 10 μm. **f** Quantification of telomeric MiDAS foci in metaphase cell treated as shown in (**d**).

In each quantification, data are presented as mean +/- s.e.m of three independent experiments. In total, 150 cells were analyzed for each condition (n = 150). *P* values were calculated with a two-tailed non-parametric Mann–Whitney test. **g** Telomere fusion blot showing telomere fusion amplicons from RPE1 wild-type (WT) cells transfected with control siRNA (Con) or siRNA targeting *POLD1* (*D1*), *POLE1* (*E1*) or *POLQ* (*Q*). The cells were also treated with a WEE1 inhibitor following nucleofection (ins insertion, del deletion). This experiment was repeated independently with similar results for four times**. h** Bar chart comparing telomere fusion molecules with complex insertion. Data plotted are means +/- s.e.m (n = 4 biological replicates). *P* values were obtained using Student's t-test (unpaired two-tailed, equal variance). **i** Scatter plot showing the size of individual DNA insertions from four biological replicates (n = 8880, 3889, 2722, 4260, 4027, 957, 1006, 989 from left to right, dotted line = median). *P* values were obtained using two-tailed Mann–Whitney test. Source data are provided as a Source data file.

mechanism to amplify actively transcribed genes or regulatory regions to drive cancer progression[15–17]. In conclusion, our data uncover mitotic MM-BIR as a key driver of CCRs, with important implications for the origin of cancers and congenital disorders.

## Methods
### Cell culture and analysis
The cell lines RPE1-hTERT (human retinal pigment epithelial cell line, CRL-4000), HCT116 (human colorectal carcinoma cell line, CCL-247) and U2OS (human osteosarcoma cell line, HTB-96) were obtained from the American Type Culture Collection (ATCC). The fibroblast MRC5 (84101801, CCL171) was obtained from European Collection of Authenticated Cell Cultures, HCA2 from James Smith (Houston, USA), IMR90 (CCL186) and WI38 (CCL75) from Coriell Institute Cell Repository. RPE1-hTERT were grown in DMEM/F12 medium (11504436, Fisher Scientific) supplemented with 1% penicillin/streptomycin, 10% foetal calf serum (FCS) and 2 mM L-glutamine. HCT116 were cultured in McCoy's 5 A medium (26600080, Thermo Fisher Scientific) supplemented with 1% penicillin/streptomycin, 10% FCS and 2 mM L-glutamine. HCT116 and RPE1 KO cell lines have been described[25,26,73]. IMR90 and Wi38 lines were cultured in Eagle's minimum essential medium (EMEM) supplemented with 15% FCS, 2× non-essential amino acids (NEAA), 26 mM Hepes, 1% penicillin/streptomycin and 2 mM L-glutamine. MRC-5 and HCA2 lines were cultured in Dulbecco's Modified Eagle Medium (DMEM) supplemented with 10% FCS, 1% penicillin/streptomycin and 2 mM L-glutamine. All cell lines with E6E7 infections were cultured in the presence of G418 (4 μg/ml). U2OS was cultured in Dulbecco's modified Eagle's medium (DMEM/Glutamax, Thermo Fisher Scientific) supplemented with 10% FBS (Thermo Fisher Scientific) and 1% penicillin/streptomycin (Thermo Fisher Scientific), at 37 °C in a humidified incubator with 5% CO$_2$. Cells were regularly tested for mycoplasma (MycoAlert, Lonza) and shown to be negative.

For telomere crisis experiments, human fibroblasts were forced to express human papilloma virus (HPV)16 E6E7 oncoproteins using amphotropic retroviral vectors, which abrogate the function of p53 and pRb, respectively, thus allowing the cells to bypass senescence and continue to divide to crisis[23]. These cells were thawed at pre-crisis stage and grown for 14 days (MRC5), 38 days (HCA2), 57 days (MRC5) and 74 days (IMR90) before cells were harvested for telomere fusion analysis. For TALEN-induced DSB experiments, HCT116 and RPE1 cell lines were nucleofected with TALEN plasmids and harvested 2 days later for telomere fusion analysis. Cell cycle analyses were performed by using a NucleoCounter NC-3000 machine (Chemometec). Inhibitors used were WEE1i (Adavosertib, HY-10993-5mg, Insight Biotech, 1 μM), CHK1i (Rabusertib, HY-14720-5 mg, Insight Biotech, 10 μM), ATRi (AZD6738, B6007-APE-5mg, Stratech Scientific, 5 μM), POLQi (ART558, HY-141520-5 mg, Insight Biotech, 10 μM), POLQi (novobiocin, B1992-APE-50 mg, Stratech Scientific, 150 μM), ATMi (KU-60019, A8336-APE-10mg, Stratech Scientific, 10 μM), LIG1i (L82-G17, HY-148161-5 mg,

Cambridge Bioscience, 50 μM) and nocodazole (M1404, Merck, 50 ng/ml).

### Generation of knockout cell lines by CRISPR/Cas9
To generate a *PIF1* KO RPE1 hTERT cell line, an antisense-stranded sgRNA targeting exon 2 of PIF1 5′-CUGAUGGUGACGAAGUCGCG-3′ was ordered from Synthego. Briefly, 100 pmol Synthego PIF1 exon 2 sgRNA along with 1 μg CleanCap 3XNLS Cas9 mRNA (TriLink), was electroporated into RPE1 hTERT cells using a Neon electroporator (Invitrogen) and allowed to recover for 2 days, at which time a portion of the targeted population, as well as an unedited population, was collected and used to make DNA samples for PCR using a Phire Tissue Direct PCR kit to confirm CRISPR/Cas9 editing of *PIF1* exon 2. PCR was performed using two primers that span exon 2 of *PIF1*, *PIF1* exon 2 TIDE ScrF 5′-ATATGAGGACTCGGAGCTGC-3′ and *PIF1* exon 2 TIDE ScrR 5′-TTAAGACGGCGGTTATGCGA-3′, to produce a 600 bp amplicon by PCR. Resulting amplicons from both populations were Sanger sequenced (Genewiz). ICE analysis (https://ice.editco.bio/#/) determined by comparing the trace files from unedited vs CRISPR-edited populations showed a Cas9-mediated indel frequency of 87% with a 1 bp insertion as the predominate indel within the edited population. Single-cell cloning of the population resulted in the isolation of KO clones. Clone 3, containing a homozygous out-of-frame G insertion at the CRISPR cut site was used for all further characterisations.

To generate a *POLD3* KO RPE1 hTERT cell line, an antisense-stranded sgRNA targeting exon 6 of POLD3 (5′- AGGUGGGCCAUGACCAUUGG-3′) was ordered from Synthego. Briefly, 100 pmol Synthego POLD3 exon 6 sgRNA along with 1 μg CleanCap 3XNLS Cas9 mRNA (TriLink), was electroporated into RPE1 hTERT cells using a Neon electroporator (Invitrogen) and allowed to recover for 2 days, at which time a portion of the targeted population, as well as an unedited population, was collected and used to make DNA samples for PCR using a Phire Tissue Direct PCR kit to confirm CRISPR/Cas9 editing of *POLD3* exon 6. PCR was performed using two primers that span exon 6 of *POLD3*, *POLD3* exon 6 TIDE ScrF 5′-AGTATTGTGTGGAAGAAGCAAGA-3′ and *POLD3* exon 6 TIDE ScrR 5′-AGTTACCCACTCTGAGCTGC-3′, to produce a 450 bp amplicon by PCR. Resulting amplicons from both populations were Sanger sequenced (Genewiz). ICE analysis (https://ice.editco.bio/#/) determined by comparing the trace files from unedited vs CRISPR-edited populations showed a Cas9-mediated indel frequency of 58% with a 1 bp insertion as the predominate indel within the edited population. Single cell cloning of the population failed to result in any homozygous KO clones with nearly all clones retaining a wild type (WT) unedited allele. One heterozygous clone, #3, containing one WT allele and one knockout allele generated by an out-of-frame T insertion at the CRISPR cut site, was further retargeted to attempt to find a homozygous KO clone. Further sequencing of 17 retargeted clones yielded no homozygous KO clones, suggesting that *POLD3* is an essential gene in RPE1 hTERT cells. The heterozygous *POLD3* clone 3 was used for all further characterisations.

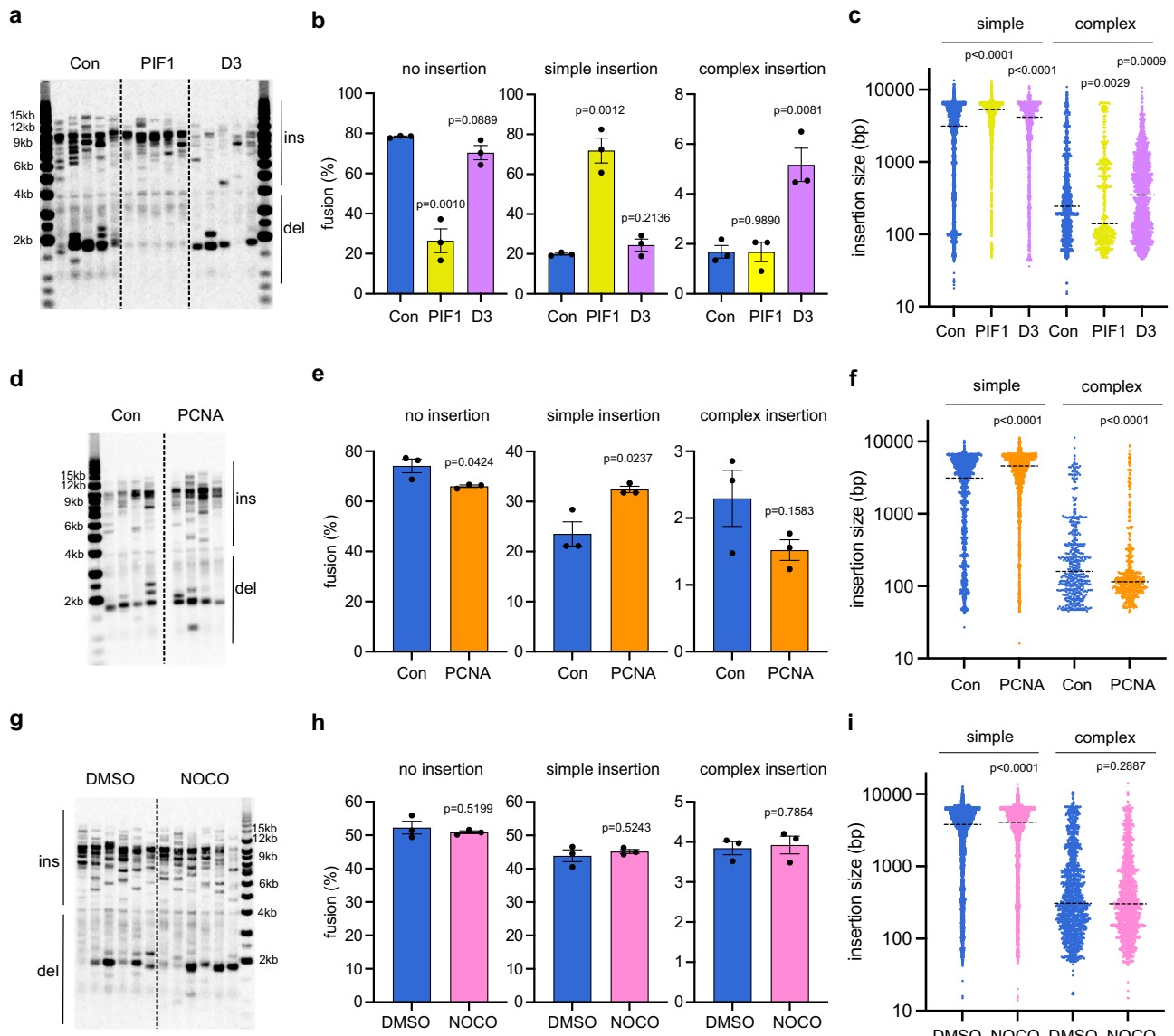

**Fig. 6 | Mitotic MM-BIR is regulated by processivity factors essential for BIR and represent a novel source of CCRs. a** Telomere fusion blot showing telomere fusion amplicons from RPE1 WT cells transfected with control siRNA (Con), *PIF1* KO cells transfected with *PIF1* siRNA (PIF1) and *POLD3* heterozygote cells transfected with *POLD3* siRNA (D3) (ins insertion, del deletion). This experiment was repeated independently with similar results for three times. **b** Bar chart comparing telomere fusion molecules. Data plotted are means +/- s.e.m (n = 3 biological replicates). *P* values were obtained as described in Fig. 3b. **c** Scatter plot showing the size of individual DNA insertions from three biological replicates (n = 5531, 4560, 1564, 1481, 454, 1880 from left to right, dotted line = median). *P* values were obtained using two-tailed Mann–Whitney test. **d** Telomere fusion blot showing telomere fusion molecules from RPE1 WT cells transfected with control siRNA (Con) or *PCNA* siRNA (PCNA) (ins insertion, del deletion). This experiment was repeated independently with similar results for three times. **e** Bar chart comparing telomere fusion molecules with no insertion, simple (1 or 2) insertion or complex (3 or more)

insertion. Data plotted are means +/- s.e.m (n = 3 biological replicates). *P* values were obtained as described in Fig. 3b. **f** Scatter plot showing the size of individual DNA insertions from three biological replicates (n = 1478, 1514, 480, 510 from left to right, dotted line = median). *P* values were obtained using two-tailed Mann–Whitney test. **g** Telomere fusion blot showing telomere fusion molecules from RPE1 WT cells treated with a WEE1 inhibitor together with DMSO or nocodazole (NOCO) (ins insertion, del deletion). This experiment was repeated independently with similar results for three times. **h** Bar chart comparing telomere fusion molecules. Data plotted are means +/- s.e.m (n = 3 biological replicates). *P* values were obtained as described in Fig. 3b. DMSO controls were the same as in Supplementary Fig. 8a–d. **i** Scatter plot showing the size of individual DNA insertions from three biological replicates (n = 5649, 4726, 1522, 1340 from left to right, dotted line = median). DMSO controls were the same as in Supplementary Fig. 8a–d. *P* values were obtained using two-tailed Mann–Whitney test. Source data are provided as a Source data file.

## Plasmids, siRNA and transfection

TALENs plasmids have been described[26,27]. siRNAs used were siGENOME SMART pool siRNAs purchased from Dharmacon, including siControl (D-001206-14-05), siPOLD1 (M-019687-02), siPOLE1 (M-020132-01), siPOLQ (M-015180-01), siPCNA (M-003289-02), siPIF1 (M-014564-02) and siPOLD3[74] (GAUGCUGUAUGAUUAAUGUUtt, from Eurofins). siRNA were transfected into RPE1-hTERT cells using Dharmafect 4 (Dharmacon) according to the manufacturer's protocol.

TALEN plasmids were nucleofected into cells using a 4D nucleofector X unit (Lonza) and SE cell line kit (program DS150 for RPE1-hTERT and DS138 for HCT116) according to manufacturer's protocol.

## RT-qPCR

RNA was isolated using an innuPREP RNA Mini Kit (BM-845-KS-2040050, Analytik Jena) according to manufacturer's protocol, and reverse transcribed using a High-Capacity cDNA Reverse Transcription

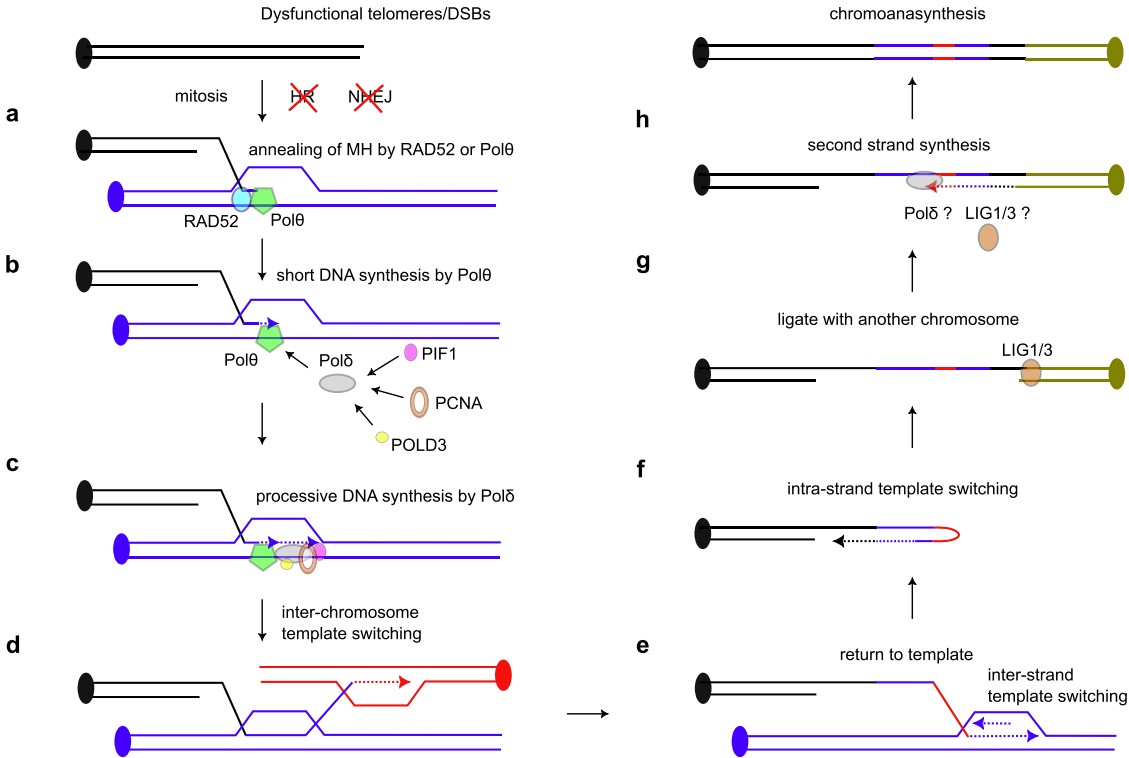

**Fig. 7 | A model showing how mitotic MM-BIR contributes to the formation of chromoanasynthesis. a, b** At dysfunctional telomeres or DNA double-strand breaks (DSBs), DNA ends utilise the strand annealing activities of RAD52 and Polθ to bind to exposed DNA with microhomologies in mitosis and initiate end bridging DNA synthesis by Polθ. **c** The DNA synthesis initiated is short but become more processive following polymerase switching to Polδ and the recruitment of processivity factors POLD3, PIF1 and PCNA. **d–f** MM-BIR initiated is highly unstable, leading to frequent inter-chromosomal, inter-strand, or intra-strand/fold-back template switching, and this process appears to continue until the DNA is ligated with another DNA end or stabilised by telomere addition. **g, h** LIG1/3 is required at the final telomere ligation step and could also contribute to joining discontinuous DNA synthesised on the second strand by Polδ.

Kit (4368814, Thermo Fisher Scientific). cDNAs were subjected to quantitative PCR using PowerUp™ SYBR™ Green Master Mix (A25780, Thermo Fisher Scientific). qPCR was performed in a ViiA7 Real-Time PCR machine (Thermo Fisher Scientific) using the following conditions, Step 1: 95 °C for 10 min (1 cycle), and Step 2: 95 °C for 15 s, 60 °C for 1 min (40 cycles). PCR Primers used were GAPDH: CCGCATCTTCTTTTGCGTCG (forward), GCCCAATACGACCAAATCCGT (reverse), POLD1: GAGCCCCTCAAAGGGGTGAG (forward), GTAGTACCTGGGAGGGGAGC (reverse), POLE1: GACCAGCATGCCTGTGTACTG (forward), CGATCTGTTCCATGAAGACCTGG (reverse), POLQ: AAATGCTCAGAGAGCCAGGG (forward), GCCTTCCGGGCACTTTTGAA (reverse), PIF1: CGAGCCTAGCACAGAAGCC (forward), CCCAGGATTCGCTTTAGCAG (reverse), POLD3: AACCAGGCCAAACAGATGCT (forward), GCAACCTTGTGGCAGGAATG (reverse) and PCNA: CTAGCCTGACAAATGCTTGC (forward), AGGAAAGTCTAGCTGGTTTCG (reverse). All PCR primers used were purchased from Eurofins.

### Western blot

Western blot analyses were performed according to a previous study[26]. Briefly, cells were resuspended in lysis buffer (150 mM NaCl, 50 mM Tris HCl, 5 mM EDTA, 1% NP40, 3 mM PMSF, 1/100 protease inhibitor cocktail III [Calbiochem 539134] and 1/100 phosphatase inhibitor cocktail II [Calbiochem 524625]) and left on ice for 5 min. Following centrifugation at 20,000 g for 30 min, whole cell extracts were removed and quantified using Bradford assay. Twenty to thirty micrograms of protein extract were resolved by Mini-PROTEAN TGX™ precast protein gels (456-1026, Biorad) and transferred to PVDF membranes (Millipore) before probing with the following antibodies: anti-GAPDH rabbit monoclonal antibody (2118, Cell Signaling Technology, RRID:AB_561053, 1:1000) and anti-POLD3 rabbit monoclonal antibody (A301-244A, Cambridge Bioscience, RRID:AB_890596, 1:1000).

### Telomere fusion PCR and Southern blotting

Single-molecule telomere fusion PCR assays were performed as described[23]. Briefly, 50 ng of phenol/chloroform extracted DNA were subjected to PCR using the following primers: 21q1 (CTTGGTGTCGAGAGAGGTAG), 17p6 (GGCTGAACTATAGCCTCTGC) and XpYpM (ACCAGGTTTTCCAGTGTGTT). Typically, four to six reactions containing 0.5 μM PCR primers and 0.5 U of a 10:1 mixture of Taq (Thermo Fisher Scientific) and Pwo polymerase (Roche) were set up and the PCR amplicons were resolved overnight at 4 °C by TAE agarose gel (0.5%) electrophoresis. Fusion bands were detected using a P33-labelled telomere adjacent probe following Southern blotting, and quantified using Image J.

### MiDAS analysis

To induce replication stress, U2OS cells were treated with 0.3 μM APH for the duration specified in the flow chart in Fig. 4I. For MiDAS analysis, asynchronous cells were arrested in the G2 phase using the CDK1 specific inhibitor RO3306 (6 μM; APExBIO), as described[75]. Cells were then rinsed three times with pre-warmed (37 °C) culture medium within 5 min, and subsequently released into pre-warmed (37 °C) culture medium containing EdU (20 μM; ThermoFisher Scientific) for 30 min. Prometaphase cells were harvested by mitotic shake-off for immunofluorescence analysis as described[54]. For treatment with POLQ inhibitor ART558 (MedChemExpress, Cat no. HY-141520), 20 μM of this inhibitor was added to the cells according to the flow chart in Fig. 4I.

## Cell treatment for MiDAS analysis combined with telomere detection on metaphase chromosomes

To induce replication stress in S phase and of POLQ inhibition at early M phase, U2OS cells were treated with 0.4 µM APH (Sigma) and 20 µM ART558 (MedChemExpress) for the duration specified in the flow chart in Fig. 5d. For MiDAS analysis combined with telomere FISH analysis, asynchronous cells were arrested in the G2 phase using the CDK1 specific inhibitor RO3306 (6 µM; APExBIO)[75]. Cells were then washed with pre-warmed (37 °C) PBS (ThermoFisher Scientific) and subsequently released into pre-warmed (37 °C) culture medium containing EdU (20 µM; ThermoFisher Scientific) and colcemid (100 ng/ml; ThermoFisher Scientific) for 1 h, which allowed cells to progress from G2 phase into metaphase. Mitotic cells were harvested by mitotic shake-off, swollen in pre-warmed (37 °C) hypotonic solution (10 mM Tris-HCl, pH 7.5, 10 mM NaCl, 5 mM MgCl2) for 20 min at 37 °C, and then fixed with methanol:acetic acid 3:1. The metaphase chromosome spreads were prepared by dropping fixed and swollen metaphase cells onto glass slides.

## Telomere fluorescence in situ hybridisation (FISH) and EdU detection

After metaphase chromosomes are fixed onto the slides, a FITC-labelled PNA telomere probe (Panagene; F1009-5) was used for telomeric FISH according to the manufacturer's protocol with the following modifications: 1× blocking reagent (11096176001, Roche) was added into the hybridisation buffer, probe incubation was performed at 37 °C for at least 2 h, and the slides containing fixed chromosomes were incubated in Washing Solution I (1× PBS with 0.1% Tween 20; pre-warmed at 57 °C) for two times of 15 min. At the end of the telomere FISH procedure, cells were re-fixed with 4% formaldehyde in 1× PBS for 4 min at room temperature, and EdU detection was carried out using Click-iT® Plus EdU Alexa Fluor® 594 Imaging Kit (C10339 Thermo Fisher Scientific). Slides containing fixed chromosomes were incubated with the reaction mix for 1 h at room temperature in the dark and were then washed in 1× PBS containing 3% BSA in and 0.5% Triton-X100 for three times (20 min each time). Slides were then mounted using Vectashield mounting medium with DAPI (Vector Labs) for microscopic analysis. Images were captured using an Olympus BX63 microscope and analyzed using CellSens (Olympus).

## Metaphase plate analysis

The growing IMR90 and HCT116 cells were incubated with 30 ng/mL colcemid (KaryoMAX, Invitrogen) for 4 h to induce metaphase arrest, treated with hypotonic buffer containing 75 mM KCl, and fixed in methanol: acetic acid (3:1). The cells were then dropped onto cold glass slides and mounted in DAPI-containing medium. Digital images were acquired using a Zeiss Apotome Axio Observer fluorescence microscope and analysed with ImageJ.

## Proximity ligation assay (PLA)

The proximity ligation assay (PLA) was performed using the Duolink PLA kit (DUO92101, Sigma) according to the manufacturer's instructions. Briefly, cells were fixed with 4% formaldehyde for 15 min and permeabilized with 0.3% Triton X-100 for 10 min. After blocking for 1 h at 37 °C, slides were incubated overnight at 4 °C with primary antibodies against TRF1 (TRF-78, ab10579, Abcam, RRID:AB_2201461) and γH2AX (2577S, Cell Signaling Technology, RRID:AB_2118010), each diluted 1:50 in blocking buffer. Slides were then washed with buffer A and incubated with the ligation solution for 30 min, followed by additional washes with buffer A. Signal amplification was performed using the amplification–polymerase solution for 100 min at 37 °C. After washing with buffer B, slides were mounted using Duolink in situ mounting medium containing DAPI. PLA signals were visualised as distinct fluorescent puncta using a Zeiss LSM800 confocal laser scanning microscope equipped with a 40× oil-immersion objective. Images were analysed using ImageJ software.

## Nanopore sequencing

Telomere fusion PCR was performed as described above and prepared for nanopore sequencing by following the Amplicons by Ligation protocol (SQK-LSK110, Oxford Nanopore Technologies) with the following modification. Briefly, 48 PCR reactions were pooled and purified by sodium acetate/ethanol precipitation followed by AMPure XP Bead cleanup (Beckman). The amplicons were ligated to a nanopore sequencing adaptor using quick T4 ligase (NEB), washed with Long Fragment buffer (LFB) and resuspended in Elution Buffer (EB). Barcoding of PCR amplicon samples for multiplex run were performed using Native Barcoding Kit 24 V14 (SQK-NBD114.24). Libraries so prepared were subsequently loaded onto MinION flow cells (FLO-MIN106D, FLO-MIN114) or Flongle Flow cells (FLO-FLG001) for nanopore sequencing.

## Fusion-seq long-read analysis pipeline (FSLR)

**Overview.** Fslr (fusion-seq long-read, available from https://github.com/kcleal/fslr) is a command line application written in Python that was developed to automate the mapping and characterisation of amplicon sequencing data generated using Oxford Nanopore Technologies R9/10 flow cells. Fslr is designed to work with any long-read amplicon data, including single or multi-primer amplicon related experiments as employed in this study.

The pipeline consists of the following main steps:
1. Read filtering to remove spurious sequences
2. Primer identification
3. Mapping input reads to the genome
4. Read clustering to identify events with multiple supporting reads

The main outputs of the pipeline include BAM files of aligned input reads and clustered sequences, along with BED files providing human-readable summaries of the alignments of reads and clusters. The bed files were analysed manually to fully characterise telomere fusion events.

The main steps of the FSLR pipeline are detailed below.

**Read filtering.** Base-called reads in FASTQ format are first filtered to remove reads harbouring probable sequencing artefacts[76], including long stretches of very low complexity bases and concatemers.

To identify runs of low complexity bases, reads are first labelled using tantan[77] which labels low complexity repeat tracks as lowercase letters. Next, blocks of low complexity in the read with length >150 bp were further assessed. To keep any reads harbouring telomere repeat arrays, we ignored repeat tracts comprised of >30% telomere repeat motifs (assessed by kmer analysis)[78]. For the remaining blocks, the repetitiveness of the sequence was quantified according to the algorithm as reported[79] (giving a score between 0 and 1 for low to high repetitiveness, respectively). If a read contained a block of sequence with a repetitiveness score >0.3, then the read was deemed to contain a sequencing artefact and was discarded. A score of 0.3 was chosen as only blocks with very low complexity fall into this category and acted as a stringent filter for potential sequencing artefacts. Although the reference genome is known to be harbour microsatellite repeats that would be removed by this filtering, we estimate that plausible sequencing artefacts occurred in our data with a frequency of around 2.6–7.5% of sequencing reads, thus necessitating stringent filtering.

Next, concatemers were identified and by aligning PCR primer sequences to the middle portion of a read using the SciKit-Bio package[80]. If a primer was identified, then the read was discarded.

**Primer identification.** Sequences in FASTQ format are labelled with the names of PCR primers. This was achieved by aligning primer sequences to the ends of reads using the SciKit-Bio package[80]. As PCR primers are generally quite short (18–26 bp), we found that using the primer sequence alone led to a loss of sensitivity and precision for

primer identification as the very ends of sequencing reads often harboured mismatches or short indels that hampered alignment. Therefore, a short stretch of the reference genome corresponding to the expected alignment location was appended to the primer sequence (this is termed the 'primer_alignment_target'), totalling 68 bp in length. This extended sequence was used to find primer matches on the read for increased precision and sensitivity when primer labelling. A primer match was identified by aligning the primer_alignment_target sequence to the read ends. If an alignment score of ≥0.4 * maximum_alignment_score was identified, then the read was labelled with the primer site and trimmed to the start of the alignment. Reads-ends without an identified primer are labelled as 'False'. In fslr, the primer sequence, 'primer_alignment_target' sequences, and alignment score threshold can be modified by the user.

**Mapping.** Reads are aligned to the reference genome using bwa mem[81], using custom settings to increase the sensitivity, setting the following options "-c 1000 -A2 -B3 -O5 -E2 -T0 -L0 -D 0.25 -r 1.25 -d 200 -k 11 -a". The -a flag is used to output all candidate alignments, which are then passed into dodi[22], which is a program for choosing an optimal set of spanning alignments. Output alignments are written to a BAM file using samtools[82].

A BED file is generated from these mappings, keeping information such as the query_start and query_end locations of each alignment in addition to mapping information.

**Clustering.** The BED file generated from the mapping step serves as input for the clustering workflow. Clustering is necessary to group multiple reads together into unique events for downstream analysis.

Clustering is conducted using a graph-based approach. The first step involves selecting reads containing complex fusions, defined as reads with three are more alignments (simple head-to-head telomere fusions are expected to yield only two alignments). Additionally, the first and last alignments are ignored for clustering as these correspond to the subtelomere locations amplified by PCR and do not provide much information for clustering purposes. Instead, clustering is performed on the inserted sequences between the subtelomeres.

Using the genomic coordinates (chromosome name, start and end position) of these insertions, fslr constructs interval trees for each chromosome, utilising the sorted-intersect package (https://github.com/kcleal/superintervals), allowing for efficient identification of overlapping intervals. To identify similar reads, fslr retrieves a list of reads that have at least one sufficiently overlapping alignment (80% overlap). These candidate reads are then further filtered by comparing the number of insertions and the query length between pairs of reads. If the number of insertions differs by 25% or the length of the query differs by >4%, then they are not further assessed. For those pairs that meet these criteria, the Jaccard similarity index is computed:

$$J(A, B) = |A \cap B| |A \cup B| J A, B = A \cap B A \cup B$$

Where A and B represent the sets of insertion alignments between two candidate reads.

If the Jaccard index for a pair of reads exceeds a user-specified threshold, then reads are added as nodes to a graph, with an edge established between them. By default, a dynamic Jaccard threshold is used depending on the number of alignments in the set. For set sizes ranging from 1 to 6, the following Jaccard thresholds are used 1, 1, 0.66, 0.66, 0.66 and 0.5. For events with more than 6 alignments, a threshold of 0.5 is used. This ensures a higher degree of precision when comparing sets with fewer alignments and increases sensitivity for sets with many alignments.

The graph is constructed using the NetworkX package[83]. To efficiently manage large datasets, the graph is built as a sparse graph, where each node is permitted a maximum of 10 edges. This approach

reduces computational overhead while still allowing effective clustering. Clusters are defined as connected components within the graph, assigning a cluster ID to all reads within the component. Nodes without edges are singletons and are assigned a unique clusters ID. Among the reads in a cluster a representative sequence is chosen based on highest average alignment score with respect the number of aligned bases. The output of this clustering process includes two BED files: one containing all reads along with information about their respective clusters, and another containing singleton reads only.

**Validation of fslr using simulated data.** To validate the performance of fslr, we developed a simulation framework to generate synthetic complex telomere fusion events. This approach allowed us to assess the accuracy of the pipeline under controlled conditions where the ground truth of fusion events was known. The simulation software is available online (www.github.com/kcleal/BadReadAmplicon) as a fork of the BadRead long-read simulator (CITE).

**Generation of synthetic fusion events.** Synthetic fusion events (n = 3000) were generated by combining sub-telomere sequences with randomly inserted genomic sequences. These insertions were selected from random chromosomal positions from the reference genome, with their lengths drawn from a gamma distribution (mean = 900 bp, standard deviation = 1600 bp). The number of insertions per fusion event followed a Poisson distribution with a mean of 5 insertions. For each insertion, both the genomic position and strand orientation were randomly determined.

To accurately model the amplification patterns observed in experimental data, we generated multiple FASTA files containing unique sequences with varying levels of amplification. The amplification levels were determined using a gamma distribution (mean = 15 reads, standard deviation = 35 reads).

**Simulation of sequencing reads.** Sequencing reads were simulated using the Badread simulator with default parameters. The quantity parameter was adjusted according to the amplification distribution described above. Each simulated read was assigned a unique identifier encoding the fusion structure, including the number of insertions and their precise genomic coordinates.

**Benchmarking alignment accuracy.** The simulated reads were processed through fslr using default parameters. Performance was evaluated by comparing the pipeline's output against the known ground truth of the simulated events. We assessed three key metrics:

1. True positives (TP): Correctly identified insertions, allowing for a 50 bp tolerance in start and end positions
2. False positives (FP): Reported insertions that did not match the simulated events
3. False negatives (FN): Simulated insertions that were not detected by the pipeline

From these measurements, we calculated standard performance metrics (Supplementary Fig. 1f):
- Precision = TP / (TP + FP)
- Recall = TP / (TP + FN)
- F1-score = 2 × TP / (2 × TP + FP + FN)

Alignment accuracy was further assessed by comparing the expected number of insertions per read against the number of mappings identified by fslr (Supplementary Fig. 1e).

**Benchmarking clustering accuracy.** A similar approach was used to evaluate the pipeline's clustering performance (Supplementary Fig. 1f). We assessed TP, FP and FN metrics by comparing the pipeline's clustering output to the known ground truth:

- TP: Intersection between identified and real clusters
- FN: Members of the real cluster that are absent in the identified cluster intersection
- FP: Members of the identified cluster that are not present in the real cluster intersection

We calculated standard performance metrics - precision, recall and F1-score – using the same method.

Clustering accuracy was further assessed by comparing the expected number of reads in a cluster against the number of reads in clusters identified by the pipeline.

### Statistical analysis

Statistical analyses were performed using Microsoft Excel and GraphPad Prism. Genome browser plots were generated by GW. All data generated were taken from distinct samples. The number of repeats (n) are indicated in the figure legends.

### Data collection and analysis

Data were collected using the following softwares: Typhoon FLA 9500 v1.1 (GE Healthcare), minKNOW v 23.07.15 (Oxford Nanopore Technologies), NC-3000™ NucleoView v2.1 (Chemometec), QuantStudio v1.3 Software (Thermo Fisher Scientific) and Zeiss Zen v2.6, blue edition (Zeiss). Data analyses were performed using the following softwares: Microsoft Excel v 16.92, GraphPad Prism v 10.2.3, FSLR v 0.3.9, GW v 1.1.1, Floreada (https://floreada.io/) v3/30/25, QuantStudio v1.3 Software (Thermo Fisher Scientific) and Image J v1.54.

### Reporting summary

Further information on research design is available in the Nature Portfolio Reporting Summary linked to this article.

## Data availability

The raw nanopore sequencing data generated in this study have been deposited in the SRA database under accession code PRJNA1194528. All analysed data are included in this paper. All materials are available upon request from the corresponding authors. Source data are provided with this paper.

## Code availability

FSLR is available at GitHub (https://github.com/kcleal/fslr) and GW[84] at GitHub (https://github.com/kcleal/gw).

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

## Acknowledgements

We thank members of the Baird, Ngo, Liu and Hendrickson/Bielinsky laboratories for help and advice. We also thank Ian Hickson (University of Copenhagen, Denmark) and Mitch McVey (Tufts University, USA) for helpful comments on the manuscript and Mitch McVey for suggesting the possible model of POLQ-induced 'cruciform trapped' DNA replication. This work was funded by the following grants: Cancer Research UK programme award C17199/A29202 (DMB), National Cancer Institute CA266254 (EAH), National Institutes of Aging AG077174 (EAH), Wellcome Trust Career Development Award 226510/Z/22/Z (GN), Danish National Research Foundation DNRF115 (YL), European Union H2020/Marie Skłodowska-Curie Actions; Antihelix 859853 (YL), Danish Independent Research Fund 1030-00180B (YL) and NEYE Foundation 122479 (YL).

## Author contributions

G.N. designed experiments, performed experiments, analysed data and wrote the manuscript; K.C. developed FSLR, provided bioinformatics expertise and edited the manuscript; S.S. provided telomere fusion data from crisis cells; V.M. developed clustering algorithms for telomere fusion data; S.A.K. provided P.L.A. and metaphase spread data; J.W.G. assisted with experimental procedures; S.B. performed MiDAS experiments; B.R. constructed many of the genetically modified cell lines used in this study; Y.L. supervised MiDAS study and edited the manuscript; E.H. provided knockout cell lines and edited the manuscript; D.B. supervised the study and edited the manuscript.

## Competing interests

The authors declare no competing interests.
