## [Transparent Peer Review file · Nature Communications]

Mitotic microhomology-mediated break-induced replication promotes chromoanagenesis

Corresponding Author: Dr Greg Ngo

Version 0:

Reviewer comments:

Reviewer #1

(Remarks to the Author)

In this manuscript, Ngo and coworkers use Oxford Nanopore Technologies following PCR amplification of molecules resulting from telomere fusions to characterize recombination and repair events that occur when cells experience telomere crisis or chromosomal damage at the subtelomeric region (a DSB induced by TALENs). A detailed analysis of a substantial collection of events is presented, revealing a range of outcomes: many simple (joining of two chromosomes at the telomere) and a few highly complex. The authors employ gene knockout cells to determine the genetic requirements for these events, demonstrating that simple outcomes are partially dependent on NHEJ, while complex ones are partially dependent on MMEJ/Polymerase theta (I find the conclusion that LIG3 knockout does not affect outcomes to be an insufficient argument for dismissing MMEJ, as LIG3 and LIG1 can act redundantly in MMEJ).

I have no major concerns regarding the quality of the data or its presentation in the figures. The manuscript provides a detailed description that offers interesting insights into the dynamics of DNA repair when cells possess recombinogenic chromosomal ends and when chromosomal fusions are selected for.

However, my primary objection lies in the interpretation presented in the title, abstract, and throughout the text. The data is framed as evidence for MM-BIR, yet I believe MM-BIR represents one possible model to explain some of the findings rather than a demonstrated mechanism. The manuscript reports on outcomes and the genes involved, but how these outcomes arise remains speculative. There is no direct evidence that recombinogenic molecules “invade” an intact chromosome and use 2–3 bp microhomologies to initiate DNA synthesis for a few hundred to a few thousand base pairs (a limitation of the PCR methodology) before jumping to another genomic region. While this is one possible explanation, I am not convinced that alternative mechanisms—such as DNA breaks, fragmentation, and NHEJ—do not account for at least part of the events. Notably, the vast majority of TALEN-induced chromosomal break events show ectopic DNA incorporated within the fusion. Given these concerns, I cannot support publication in its current form, as I strongly believe the conclusions are not sufficiently supported by the data. For example, the title “MM-BIR promotes ...” should be revised to “MM-BIR may promote ...” to reflect the uncertainty, and that would immediately take down what could be the most interesting novelty.

In my view, there are also several observations contradictory to the overarching claim:

- The effect of LIG4: How is it that the LIG4 knockout effect in Figure 3D on complex events is not statistically significant (despite an evidently different distribution), while the LIG3 knockout effect on simple events is significant despite only a minor deviation in distribution?
- There is a clear correlation between the origin of the inserts (Figure 1F) and the chromosomes analyzed (Figure 1D). Do these inserts map close to the telomeres? Could they not be the result from BFB cycles?
- The genetic findings are inconsistent, particularly regarding POLQ. In Figure 4C, the effect of POLQ knockout only partially affects the complex outcomes. What does this imply? Could there be another POLQ-like enzyme involved in MM-BIR with similar properties?
- Similarly, in Extended Data Figure 7B and 7C, the text (line 324) states that “reduced MH size and increased the frequency of blunt junctions... [were] observed in PolQ-depleted cells.” However, these effects appear extremely marginal, despite the strong existing evidence that MH usage is a defining feature of POLQ activity. Either the knockdown is highly inefficient, or another mechanism redundantly compensates for POLQ function—but what would that be?

Thus while the manuscript contains interesting data, I believe it suffers from significant overinterpretation. Moreover, I am

unsure whether a revised version with a more cautious interpretation would constitute a strong candidate for Nature Communications, as it would be largely descriptive without providing a clear mechanistic insight.

(Remarks on code availability)

Reviewer #2

(Remarks to the Author)

Gno and Baird present a manuscript concerning BIR events at short telomeres and subtelomeres that can create highly complex rearrangements in the context of tumor development. Interestingly, pieces of DNA from all chromosomes were detected at fusions of 3 telomeres. Telomeres were sometimes in tact but deletions within subtelomeres were common prior to fusion and apparently always occurred for 21q. The authors discovered highly complex insertions at telomeres for telomerase-negative fibroblasts grown until crisis. The authors were careful to use four independent fibroblast lines, thereby ensuring that their results are not confounded by genetic background. As a nice control, the authors generated telomere fusions using talen nucleases that cut 1-2 kb from telomeres. This allows the authors to ask if fusions created as a consequence of telomere shortening are distinct from those that occur from a DSB that is very closeby in the subtelomere. There were few talen fusion events with insertions and when insertions were present these were diminished in number. However several highly complex events occurred with insertions from many different chromosomes, as seen for fusions created in the absence of telomerase. At one fusion, more than 100 insertions from 18 chromosomes occurred. The authors have identified a consistent remarkable process of fuions, promoted by inversions or simple repeats present adjacent to the origins of insertions.

The authors interrogate proteins implicated in end joining and observed a strong reduction in fusions in the absence of LIG4, suggesting NHEJ. However, complex events were not dependent on LIG4 or LIG3. Although RAD52 loss did not affect fusion frequency, it did markedly diminish the complex events, suggesting a strand invasion mechanism. The authors further find that polq fibroblasts have a diminished complex fusion events, but that the size of those that did occur increases, indicating that polq contributes to one pathway that creates complex insertions. Observations in a polq mutant background are nicely complemented by polq inhibitors that affect distinct polq activities. The authors nicely show that pold and pole contribute to insertions in complementary ways, perhaps explaining residual events that occur when polq is absent. The authors also use nocodazol to show that complex insertions occur in early mitosis. Overall, the authors present compelling evidence that mm-bir and polq are used in mitosis to create complex insertions at sites in the genome that have inversions and are prone to fork stalling when replication is perturbed.

specific points:

1. For the 21q family of telomeres, could the authors provide an estimate of the number of telomeres involved.
2. It is curious that an entire family of telomeres would almost always experience terminal deletions prior to fusion. Why would this be? Is there a subtelomere sequence that might be responsible for this instability?
3. For comparison with Talen data in Fig. 3b, it would be helpful if the authors could add a bar for the fraction of rearrangements that occur in crisis in the absence of telomerase. This would allow readers to readily assess the subtelomeric Talen DSB insertions with those that occur at telomeres that become critically short in the absence of telomerase.
4. Polq facilitates DSB repair in mitosis. To confirm this, the authors inactivated the G2/M DNA damage checkpoint (line 250). It would be helpful if the authors could explain here what happens to polq-healed DSBs if these inhibitors are used. Has polq activity been shown to be inhibited by or increased if the the G2/M DNA damage checkpoint is inhibited? This would help the reader to understand the high levels of insertion that occur when these inhibitors are use, and the relationship of these inhibitors to mitosis and the mitotic repair activity of polq.
5. 'template switching, and this process appear to continue until'; appears

(Remarks on code availability)

Reviewer #3

(Remarks to the Author)

Comments on "Mitotic Microhomology-Mediated Break-Induced Replication Promotes Chromoanasythesis"

In this manuscript, the authors describe that upon telomere crisis, chromoanasythesis—a type of complex chromosome rearrangement (CCR) characterized by numerous insertions with microhomologies at tightly clustered breakpoints due to template switching—occurs via microhomology-mediated break-induced replication (MM-BIR). Utilizing Oxford nanopore sequencing technology for single-molecule long-read DNA sequencing, the authors analyze DNA and subtelomeric DNA. They report that chromoanasythesis takes place at short telomeres and sub-telomeric double-strand DNA breaks, asserting that MM-BIR occurs during mitosis and is promoted by RAD52, Polθ, and Polδ. This analysis could shed light on how

chromosome rearrangements at the telomere region promote cancer and some congenital disorders. While I find this work interesting, I believe the data presented are premature to fully support the authors' conclusions. Below are my comments.

Major Comments:

1. **Complex Chromosomal Rearrangement Analysis:** I acknowledge that analyzing complex chromosomal rearrangements is challenging. However, the authors claim that chromoanasythesis is the primary CCR responsible for telomere fusions and aberrations during replication stress, telomere crisis, and double-strand breaks at the telomere. The authors appear to have overlooked other CCR including chromothripsis. The analysis of LIG3/LIG4 abrogation—responsible for non-homologous end joining (NHEJ)—did not have a substantial impact. As is well known, chromoanasythesis can occur alongside chromothripsis. I am not fully convinced that chromothripsis, kataegis, or other types of CCR can be disregarded concerning telomere-telomere fusion under conditions of telomere crisis. The frequencies reported in the analyzed data do not adequately exclude other CCR mechanisms. Furthermore, in the discussion section, the authors suggest that replication stress triggers chromoanasythesis, while micronuclei or chromatin bridges serve as sources of chromothripsis. This idea is intriguing, but with the provided data, I require more convincing evidence.

2. **Role of RAD52 in MM-BIR:** The authors argue that MM-BIR is responsible for chromoanasythesis and assert that RAD52 is essential for this process. However, this contradicts established knowledge; two major forms of BIR have been documented: RAD52-dependent and RAD52-independent. Many studies indicate that MM-BIR does not require RAD52. Additionally, the data in this manuscript do not convincingly demonstrate that RAD52 is critically required for the process. I encourage the authors to investigate this matter more thoroughly to ascertain whether the process is indeed MM-BIR and to exclude other forms of BIR.

3. **Analysis of Deletion Size and Microhomology:** The analysis regarding deletion sizes and microhomologies at the breakpoints does not sufficiently clarify the complexity of the CCR within telomere crisis. For instance, the presence of large deletions and a wide range in insertion sizes (from 55 bp to 3,916 bp) raises questions. Can the authors completely rule out the possibility of standard BIR in this context?

4. The authors utilized normal fibroblasts in culture that undergo telomere crisis after a finite number of cell divisions. Additionally, they employed Aphidicolin to induce replication stress. In this context, the telomere DNA damage appears to be ATR-dependent, which differs significantly from TALEN-induced double-strand breaks (DSBs). The employment of FSLR analysis to investigate subtelomere DSB-induced CCR raises questions about why the authors found the process to be ATR-independent and how two distinct types of DNA lesions provoke the same MM-BIR pathway, leading to chromoanasythesis.

5. **MM-BIR During Mitosis:** The authors assert that MM-BIR occurs during mitosis, linking it to mitotic DNA synthesis (MiDas), which is typically associated with BIR. However, the claim that ATR inhibitors had marginal effects while WEE1 and CHK1 inhibitors increased CCR raises questions. Additionally, the MiDas assay depicted in the figures does not explicitly involve telomeres, making it unclear whether the sequence analysis is relevant to the MiDas assay presented in this work. I recommend that the authors repeat experiments with the inhibitors and also employ gene depletion methods (such as siRNA, CRISPR, or shRNA) to corroborate the analysis concerning ATR, WEE1, CHK1, LIG3, LIG4, POLQ, etc. This is essential for reaching any firm conclusions.

6. **ALT Mechanism and Analysis of ALT Cell Lines:** Break-induced replication is a key mechanism for Alternative Lengthening of Telomeres (ALT). The authors have conducted their analyses using normal fibroblasts. What about the ALT cells? Many reports suggest that replication stress provokes ALT. Therefore, it would be interesting to compare the signatures of telomere sequences between normal fibroblasts undergoing telomere crisis and ALT cells. I suggest that the authors compare the frequencies of CCR, along with the insertions, deletions, and copy numbers at both the telomere and subtelomere, to provide a more comprehensive understanding of the process in the context of ALT.

Minor Comments:

1. On page 6, line 166, the authors state that the microhomologies at the junctions were approximately 51.8%. What is the significance of the remaining percentage? Is this sufficient evidence to support the claim of MM-BIR?

2. On page 7, line 200, and in Fig. 3A, the authors report that RAD52 knockout (KO) "mildly" reduced the level of insertions. I recommend a more precise characterization of the observed effects.

3. On page 9, line 274, the reference to the TMEM105 oncogene being implicated in breast cancer appears to be over-interpreted. I suggest a more cautious phrasing to avoid overstating the evidence.

4. PolQ is not specific to the mitotic phase. I recommend toning down any references to it being associated exclusively with the mitotic cell cycle.

5. On page 11, lines 337-339, the statement that "MM-BIR occurs in early mitosis to generate telomere dysfunction-induced chromoanasythesis independently of the formation of chromatin bridges or micronuclei" may be overstated, given that fibroblasts have a finite cell division cycle. Micro-nuclei result from chromosomal mis-segregation and a small number of micronuclei that do not degenerate can incorporate into the genome, leading to chromothripsis. It is unclear whether the experimental design sufficiently accounts for the number of cell divisions required for meaningful chromosomal mis-segregation. I am not sure whether the authors can definitively rule out the possibility of micronuclei formation and chromothripsis upon telomere crisis.

6. Could the authors provide information on how many times the fibroblasts divided before committing to telomere crisis?

7. In Fig. 2, the authors conducted subtelomere analysis using HCT116 cells. Could they justify the use of different cell types in this context?

8. In Fig. 3B, the differences in CCR between LIG3, LIG4, and RAD52 are not as pronounced as the authors claim. A clearer presentation of the data may be helpful.

9. In Fig. 4J, the MiDas assay with EdU should ideally be performed in conjunction with telomere-FISH (T-FISH) for enhanced clarity. Additionally, the differences noted in the results are not markedly clear.

(Remarks on code availability)

Version 1:

Reviewer comments:

Reviewer #1

(Remarks to the Author)

The impressively revised manuscript contains a substantial number of new experiments. Several of these address my initial concerns while their outcomes strengthen the manuscript, as well as increase the impact/novelty of the study. I also felt that the discussion section has improved.

I therefore support publication of the manuscript in its current form.

(and I wish to compliment the authors on such nice work!)

(Remarks on code availability)

Reviewer #2

(Remarks to the Author)

This is an outstanding manuscript that thoroughly interrogates the roles of RAD52, DNA pol theta, PIF1 PCNA and POLD in creation of complex fusion events that occur during early mitosis at shortened telomeres or at subtelomeric DSBs. These data are highly relevant to common genome rearrangements that occur during tumor development but are difficult to detect based on short read sequencing of tumor genomes. The authors have done a nice job of addressing concerns raised by the reviewers, including a number of experiments. This well written manuscript is suitable for publication in Nature Communications.

(Remarks on code availability)

Readme files are available that clearly indicate how to process their long read sequencing data sets.

Reviewer #3

(Remarks to the Author)

Comments for the revised manuscript.

Overall, the authors have adequately addressed several points raised by previous reviewers; however, there remain substantial conceptual and experimental gaps that significantly weaken the central claims of the manuscript. In its current form, despite the revision, the evidence provided is still insufficient. Although the study presents interesting observations and the topic is of high relevance, the current data does not adequately support the major conclusions of the manuscript. In my view, significant additional clarification and experimental strengthening are required before the study can be considered for publication. My major concerns are summarized below.

Major Concerns

1. Lack of direct cellular evidence demonstrating telomere crisis or telomere damage

The authors state that telomere crisis was induced either via TALEN mediated targeted telomere cutting or population doubling. However, no direct cellular or cytogenetic validation is presented to demonstrate that the cells indeed underwent telomere crisis or exhibited telomere dysfunction. To substantiate this foundational assumption, the manuscript should provide clear cellular evidence such as telomere dysfunction induced foci (TIF assay: γ H2AX or 53BP1 co localization with telomere FISH), metaphase spread analysis of chromosome end to end fusions, or telomere fragility markers. Without these data, it is difficult to interpret subsequent claims regarding telomere crisis induced genome rearrangements.

2. Insufficient explanation regarding cell type specific effects, particularly the stronger impact in HCT116

The manuscript does not adequately explain why chromoanasythesis is most pronounced in HCT116 cells while being minimal in U2OS, despite U2OS being an ALT cell line with high recombination activity, persistent replication stress, and MM-BIR competence. Given that ALT cells typically show elevated RAD52 and POLQ dependent recombination and robust

templated insertion at DSBs, the unusually low insertion frequency observed in U2OS is unexpected. This discrepancy raises the possibility that the subtelomeric TALEN cut is inefficient in U2OS or that ALT cells process subtelomeric breaks differently from HCT116. To clarify this issue, the authors should examine TALEN cutting efficiency, end resection at the break, expression or activity of MM-BIR related factors (e.g., POLQ, RAD52), and ALT specific telomere features such as APBs or telomere length heterogeneity. Without addressing these possibilities, the cell type specific differences remain unexplained, weakening the mechanistic interpretation of the results.

Min, J. et al. Mechanisms of insertions at a DNA double-strand break. *Molecular Cell* 83, 2434–2448 (2023).

3. The evidence supporting mitotic MM-BIR relies almost exclusively on MiDAS data, which are of insufficient quality. The central claim that chromoanasythesis occurs during mitosis through mitotic MM-BIR is largely based on MiDAS. However, the MiDAS imaging data are of poor quality; EdU and telomere staining appear weak or inconsistent even in control samples, making interpretation unreliable.

4. Unclear mechanistic relationship between RAD52 and Polθ

The manuscript continues to lack conceptual clarity regarding the relationship between RAD52 and Polθ in promoting chromoanasythesis. The authors simultaneously state that RAD52 is required while Polθ acts in a RAD52 independent manner, which appears contradictory without mechanistic reconciliation. It is critical to clarify whether RAD52 functions upstream for ssDNA annealing/template preparation, while Polθ drives later template switching and extension, or whether they act in parallel pathways. Without a clearer mechanistic explanation, the molecular model remains incomplete.

5. Interpretation of the POLD3 phenotype

Prior work has demonstrated that POLD3 loss induces substantial replication stress and cell death phenotypes. In your dataset, POLD3 depletion increases the frequency and tract length of complex insertions, which is interpreted as enhanced MM-BIR driven template switching. However, because POLD3 deficiency itself triggers pronounced replication stress, an alternative explanation is that the increase in complex events reflects globally elevated fork instability, rather than a direct mechanistic shift in MM-BIR dynamics. To distinguish between these possibilities, I strongly recommend performing POLD3 depletion under conditions where POLQ is simultaneously depleted, so that

MM-BIR patching is suppressed but replication stress is still induced. This would clarify whether the increase in complex insertions is truly due to MM-BIR dysregulation or secondary to replication stress.

Buzovetsky, O. et al. Role of the Pif1-PCNA Complex in Pol δ-Dependent Strand Displacement DNA Synthesis and Break-Induced Replication. *Cell Reports* 21, 1707–1714 (2017).

6. Interpretation of the PIF1 phenotype

Several studies have shown that PIF1 contributes to both replication coupled and replication independent forms of BIR. Therefore, the finding that complex insertions in your system are largely unaffected by PIF1 depletion is unexpected. This discrepancy raises the possibility that the events classified as complex insertions may not arise from a single, uniform recombination-based mechanism. Moreover, although LIG4 knockout did not produce a statistically significant reduction in complex insertions, there is a consistent trend toward decreased complex events in both the WT and LIG3 deficient backgrounds. Even if LIG4 is not directly required for complex insertions, there remains the possibility of MMEJ or SSA mediated cut and paste like insertion events, which rely on LIG3 or LIG3/RAD52, respectively. Therefore, to more rigorously define the recombination pathway responsible for complex insertions, it would be informative to examine complex insertion frequency under conditions where both POLD3 and POLQ are simultaneously depleted. Dual suppression of POLD3 and POLQ mediated BIR synthesis should more completely eliminate MM-BIR activity.

Li, S. et al. PIF1 helicase promotes break-induced replication in mammalian cells. *EMBO Journal* 40, e104509 (2021).

Wu, T. et al. Break-induced replication is activated to repair R-loop-associated double-strand breaks in SETX-deficient cells. *Cell Reports* 44, (2025).

7. ATR inhibition and cell cycle control

The manuscript concludes that ATR inhibition does not significantly alter cell cycle distribution. However, ATR inhibition causes a marked increase in simple insertions, consistent with the elevated simple insertion possibly by globally elevated DNA fragment pools that are captured by LIG4 dependent NHEJ. If ATR inhibition indeed produces widespread fork collapse and DNA fragmentation, one would mechanistically expect cell cycle perturbation, particularly at the G2/M checkpoint. It would strengthen the manuscript to clarify why ATR inhibition fails to affect cell cycle profiles despite an increase in simple insertion events.

(Remarks on code availability)

Version 2:

Reviewer comments:

Reviewer #3

(Remarks to the Author)

The authors have adequately addressed the major concerns raised during the previous round of review, including those from junior scientists. The revised manuscript shows substantial improvement in both clarity and experimental support. In particular:

The authors now provide appropriate telomere dysfunction evidence using proximity ligation-based TIF assays and

metaphase spread analyses, confirming that the experimental system indeed induces telomere damage.

The role of POLD3 in a POLQ-deficient background is now appropriately evaluated. Notably, POLD3 depletion does not increase complex insertions in the absence of POLQ, suggesting that the frequent template switching observed upon POLD3 depletion is POLQ-dependent.

The revisions also resolve earlier conceptual ambiguities. The authors now propose that RAD52 and POLQ operate in parallel, promoting microhomology annealing and POLD recruitment, respectively. Relevant textual sections have been appropriately edited to reflect this clarification.

This reviewer had requested additional experiments concerning TALEN cutting efficiency and the effects of ATR inhibition on cell-cycle profiles. While these were not addressed with new data, the authors have discussed these points in a manner that is reasonable at the current stage.

Taken together, the manuscript is now substantially strengthened and appropriate for publication in Nature Communications.

(Remarks on code availability)

To all reviewers

We thank the reviewers for their insightful and constructive comments. In response to the reviewers' comments we have undertaken further experimentation and provided new data that we believe strengthens our conclusions; we hope you agree that the manuscript has improved as a result. Our responses to the reviewers' comments and details of the amendments to the manuscript are provided below.

We would like to point out a mistake made in preparing our initial manuscript, which affected Fig. 3b and Fig.4b. This was discovered when we tried to replot these staggered graphs to make it easier to compare the level of complex insertions, as requested by reviewer 3. Due to an Excel formatting error, we found that the statistics associated with complex insertions were incorrect. The corrected statistical differences are listed below:

LIG3 KO – changed from not significant to **

LIG4 KO – remain not significant

LIG3/LIG4 DKO – changed from not significant to **

RAD52 KO – changed from ** to ***

POLQ KO – changed from * to ***

We have incorporated LIG3 and LIG1 (this new data was generated in response to Reviewer 1's comment) into our new model of MM-BIR (Fig.7).

In the new graphs (Fig. 3b and 4b) showing the level of insertion, we have also combined fusion with 1 or 2 insertions into one group called simple insertion, which simplify the analysis and make it consistent with the other figures. This new grouping does not affect our interpretation of the data compared to the original manuscript.

In addition to these insertion level plots, we have also converted all other staggered plots (junction, external DNA, cell cycle) into plots with separate bars to make it easier to see the differences.

REVIEWER COMMENTS

Reviewer #1 (Remarks to the Author):

In this manuscript, Ngo and coworkers use Oxford Nanopore Technologies following PCR amplification of molecules resulting from telomere fusions to characterize recombination and repair events that occur when cells experience telomere crisis or chromosomal damage at the subtelomeric region (a DSB induced by TALENs). A detailed analysis of a substantial collection of events is presented, revealing a range of outcomes: many simple (joining of two chromosomes at the telomere) and a few highly complex. The authors employ gene knockout cells to determine the genetic requirements for these events, demonstrating that simple outcomes are partially dependent on NHEJ, while complex ones are partially dependent on MMEJ/Polymerase theta (I find the conclusion that LIG3 knockout does not affect outcomes

to be an insufficient argument for dismissing MMEJ, as LIG3 and LIG1 can act redundantly in MMEJ).

I have no major concerns regarding the quality of the data or its presentation in the figures. The manuscript provides a detailed description that offers interesting insights into the dynamics of DNA repair when cells possess recombinogenic chromosomal ends and when chromosomal fusions are selected for.

However, my primary objection lies in the interpretation presented in the title, abstract, and throughout the text. The data is framed as evidence for MM-BIR, yet I believe MM-BIR represents one possible model to explain some of the findings rather than a demonstrated mechanism. The manuscript reports on outcomes and the genes involved, but how these outcomes arise remains speculative. There is no direct evidence that recombinogenic molecules “invade” an intact chromosome and use 2–3 bp microhomologies to initiate DNA synthesis for a few hundred to a few thousand base pairs (a limitation of the PCR methodology) before jumping to another genomic region. While this is one possible explanation, I am not convinced that alternative mechanisms—such as DNA breaks, fragmentation, and NHEJ—do not account for at least part of the events. Notably, the vast majority of TALEN-induced chromosomal break events show ectopic DNA incorporated within the fusion. Given these concerns, I cannot support publication in its current form, as I strongly believe the conclusions are not sufficiently supported by the data. For example, the title “MM-BIR promotes ...” should be revised to “MM-BIR may promote ...” to reflect the uncertainty, and that would immediately take down what could be the most interesting novelty.

We thank the reviewer for these constructive comments. To further support the involvement of MM-BIR, we have examined the requirement for PIF1 and POLD3, two processivity factors specifically required for extensive DNA synthesis during BIR. We generated PIF1 and POLD3 mutant cell lines and further knocked down these genes using siRNA. Importantly, PIF1 depletion reduced the size of complex insertions (Fig.6c), supporting the hypothesis that long insertions detected in our CCR assay are generated by BIR. This is further supported by our observation that depletion of PCNA, which is essential for PIF1 activity in BIR (Buzovetsky et. al 2017, Li et. al 2021), induced the same short insertion phenotype (Fig.6f). On the other hand, POLD3 depletion increases the number of complex insertions (Fig.6b), consistent with these CCRs being regulated by the stability of Pol δ -dependent DNA replication, further supporting the involvement of BIR.

These findings strongly strengthen our hypothesis that CCRs are generated by a replicative mechanism, further supporting evidence based on replicative signatures detected on CCRs, which includes foldback-induced DNA synthesis, template switching at inverted repeats and events containing more than 3 overlapping insertions (Figs.1 and 2). Based on these findings, we propose an exciting new model of mitotic MM-BIR pathway, which involve MMEJ proteins initiating a Pol δ -dependent BIR pathway which is regulated by PIF1, POLD3 and PCNA (Fig.7 and discussion).

Thank you for the helpful comments regarding redundancy between LIG1 and LIG3. To test this redundancy, we have performed experiment examining the effect of a LIG1 inhibitor in LIG3 KO and found an important role for LIG1/3 in the generation of complex insertions (but not simple insertions) (Fig.3i). These findings allow us to propose that simple insertions, which primarily involve ligation with TALEN plasmid, are dependent on LIG4-mediated

NHEJ, whereas complex insertions are dependent on LIG1/3, consistent with their interactions with Pol θ and Pol δ (Stroik et.al 2023). We have modified our model to incorporate the roles played by LIG1/LIG3 in MMEJ (final ligation step between telomeres) and BIR (ligation of discontinuous DNA generated during second strand synthesis, Donnianni et. al 2019) into a new model of MM-BIR (Fig.7).

We have made extensive amendments to the manuscript to describe these new findings, which are highlighted in red on pages 8, 12 and 13.

These new data also allow us to propose a more flexible model whereby MMEJ-based ligation and MM-BIR could collaborate to generate CCRs, as detected in some complex events and summarised in the new paragraph below (page 16):

Despite the involvement of mitotic MM-BIR in chromoanasythesis, our model does not completely rule out a role for MMEJ-induced ligation. Due to the presence of MMEJ factors Pol θ and Pol δ , it is possible that once DNA ends engaged in MM-BIR dissociate from their template, they can switch to an MMEJ mode and ligate with other DNA fragments. In support of this hypothesis, unique insertion patterns observed in WEE1 inhibitor treated cells show that short, fragmented DNA can be ligated into some CCR events (Supplementary Fig. 7c-f). Furthermore, most of the MM-BIR events detected here rely on MMEJ to ligate with another telomere and complete the repair process. The tight association between MM-BIR and MMEJ proteins could explain the exclusive requirement of LIG1/3, instead of LIG4, in MM-BIR. This plasticity between MM-BIR and MMEJ usage could contribute to co-occurrence of chromoanasythesis and chromothripsis^{8,13,14,17,21}. However, the detection of such events, especially megabase-scale DNA fragmentation and ligation described in chromothripsis is difficult in our assay due to the amplification limit of PCR.

Buzovetsky, O. et al. Role of the Pif1-PCNA Complex in Pol δ -Dependent Strand Displacement DNA Synthesis and Break-Induced Replication. *Cell Reports* 21, 1707–1714 (2017).

Li, S. et al. PIF1 helicase promotes break-induced replication in mammalian cells. *The EMBO Journal* 40, e104509 (2021).

Stroik, S. et al. Stepwise requirements for polymerases delta and theta in theta-mediated end joining. *Nature* 623, 836–841 (2023).

Donnianni, R. A. et al. DNA Polymerase Delta Synthesizes Both Strands during Break-Induced Replication. *Molecular Cell* 76, 371-381.e4 (2019).

In my view, there are also several observations contradictory to the overarching claim:

- The effect of LIG4: How is it that the LIG4 knockout effect in Figure 3D on complex events is not statistically significant (despite an evidently different distribution), while the LIG3 knockout effect on simple events is significant despite only a minor deviation in distribution?

This is due to log scale used in the graph. The same graph plotted in linear scale below shows that LIG3KO has a bigger increase in simple insertion size, in comparison to LIG4KO which do not show a big difference in complex insertion size.

- There is a clear correlation between the origin of the inserts (Figure 1F) and the chromosomes analyzed (Figure 1D). Do these inserts map close to the telomeres? Could they not be the result from BFB cycles?

There appears to be some correlation between the chromosomal origins of flanking DNA and insertion. To better present this data, we have added new karyotype plots to show the origin of insertions from telomere crisis or TALEN experiments (Supp Fig.1f and Supp Fig.2g). The data shows that while many insertions are mapped to non-telomeric loci, a large proportion of insertions do originate from telomeric regions.

We don't consider that this is the result of BFB cycles because we did not observe a difference in insertions in the presence of nocodazole (Fig. 6h), which prevents entry into anaphase and the occurrence of BFB. Furthermore, BFB cycle would be expected to generate insertions at random sites of broken bridges and not within fused telomeres as we have observed, moreover insertions arising from BFB usually result in characteristic stair-like duplications (Li et.al 2023); which were not observed in our complex molecules. It would also be difficult to envisage BFB generating high number of insertions which originate from different regions of the genome, some of which are overlapping (but not duplicated) insertions.

Instead, we believe that this observation could be due to DNA repair using local DNA templates, either intra-molecularly by folding back on the DNA itself, which could be driven by POLQ-mediated stem loop formation and snap-back DNA synthesis (Fijen et. al 2024, Carvajal-Maldonado et.al 2024) or inter-molecularly by using another broken/uncapped telomere that could be brought to close proximity by clustering of DNA breaks (Arnould et. al 2023).

Li, C., Chen, L., Pan, G., Zhang, W. & Li, S. C. Deciphering complex breakage-fusion-bridge genome rearrangements with Ambigram. *Nat Commun* 14, 5528 (2023).

Fijen, C. *et al.* Sequential requirements for distinct Pol theta domains during theta-mediated end joining. *Mol Cell* 84, 1460-1474 e6 (2024).

Carvajal-Maldonado, D. et al. Dynamic stem-loop extension by Pol θ and templated insertion during DNA repair. *Journal of Biological Chemistry* 300, 107461 (2024).

Arnould, C. et al. Chromatin compartmentalization regulates the response to DNA damage. *Nature* 623, 183–192 (2023).

- The genetic findings are inconsistent, particularly regarding POLQ. In Figure 4C, the effect of POLQ knockout only partially affects the complex outcomes. What does this imply? Could there be another POLQ-like enzyme involved in MM-BIR with similar properties?

We thank the reviewer for raising this point. We believe that the complex events generated in POLQ KO (Fig.4b,c) are due to redundancy in the MM-BIR pathway and indeed we provide new data showing that POLQ and RAD52 act in alternative pathways to initiate MM-BIR (Fig.4i, page 10,11). We propose that this is because these proteins exhibit similar activities in annealing DNA with microhomology and recruiting Pol δ (Bhowmick et. al 2016, Stroik et al. 2023).

Bhowmick, R., Minocherhomji, S. & Hickson, I. D. RAD52 Facilitates Mitotic DNA Synthesis Following Replication Stress. *Mol Cell* 64, 1117–1126 (2016).

Stroik, S. et al. Stepwise requirements for polymerases delta and theta in theta-mediated end joining. *Nature* 623, 836–841 (2023).

- Similarly, in Extended Data Figure 7B and 7C, the text (line 324) states that “reduced MH size and increased the frequency of blunt junctions... [were] observed in PolQ-depleted cells.” However, these effects appear extremely marginal, despite the strong existing evidence that MH usage is a defining feature of POLQ activity. Either the knockdown is highly inefficient, or another mechanism redundantly compensates for POLQ function—but what would that be?

The reduction in microhomology (MH) size may be small, but it is a consistent finding across numerous independent experiments. A recent comparable paper which analysed MH usage following POLQ depletion in human cells shows that in the absence of POLQ, DNA repair is largely channelled to another ‘genetically complex’ pathway that also favours MH, but to a lesser extent than POLQ (Carvajal-Garcia et.al 2020). This is consistent with our finding that there is high level of MH following POLQ knockout/depletion/inhibition (POLQ KO, siPOLQ, POLQ inhibitors), and that MH size is reduced in these conditions.

We have also shown that RAD52 provides a POLQ independent pathway in MM-BIR (Fig.4i, page 10,11), suggesting that MH annealing activity of RAD52 could compensate for POLQ’s function. The presence of MH in RAD52 KO treated with POLQ inhibitors could be due to incomplete inhibition of POLQ or the presence of another unidentified pathway.

Carvajal-Garcia, J. et al. Mechanistic basis for microhomology identification and genome scarring by polymerase theta. *Proc Natl Acad Sci U S A* 117, 8476–8485 (2020).

Thus while the manuscript contains interesting data, I believe it suffers from significant overinterpretation. Moreover, I am unsure whether a revised version with a more cautious interpretation would constitute a strong candidate for Nature Communications, as it would be largely descriptive without providing a clear mechanistic insight.

We are sorry the reviewer considers our findings in this way. In this manuscript, we describe for the first time, astonishingly complex mutation events, consistent with chromoanasythesis, occurring directly because of telomere dysfunction in human cells, both in cells undergoing replicative telomere crisis and following the experimental induction of telomeric DSBs. Replicative crisis is a key step that drives genetic diversity and clonal

progression to malignancy; thus, we consider that characterising the mutational impact of telomere dysfunction is significant in and of itself.

Moreover, our ability to induce these events and characterise them with a novel long read DNA sequencing approach has allowed us to investigate the genetic requirements of this elusive phenomenon in human cells. We show that chromoanasythesis is not dependent on NHEJ but generated by a BIR pathway which is initiated by the MMEJ machinery. This MM-BIR pathway requires novel and unexpected collaboration between proteins involved in MMEJ (Pol θ) and BIR (PIF1, POLD3 and Pol δ) in mitosis, which help to explain its highly mutagenic nature. We believe our findings have significant implications for understanding the origin of mutations in cancer and congenital disorders and therefore represent a strong candidate for Nature Communications.

Reviewer #2 (Remarks to the Author):

Gno and Baird present a manuscript concerning BIR events at short telomeres and subtelomeres that can create highly complex rearrangements in the context of tumor development. Interestingly, pieces of DNA from all chromosomes were detected at fusions of 3 telomeres. Telomeres were sometimes in tact but deletions within subtelomeres were common prior to fusion and apparently always occurred for 21q. The authors discovered highly complex insertions at telomeres for telomerase-negative fibroblasts grown until crisis. The authors were careful to use four independent fibroblast lines, thereby ensuring that their results are not confounded by genetic background. As a nice control, the authors generated telomere fusions using talen nucleases that cut 1-2 kb from telomeres. This allows the authors to ask if fusions created as a consequence of telomere shortening are distinct from those that occur from a DSB that is very closeby in the subtelomere. There were few talen fusion events with insertions and when insertions were present these were diminished in number. However several highly complex events occurred with insertions from many different chromosomes, as seen for fusions created in the absence of telomerase. At one fusion, more than 100 insertions from 18 chromosomes occurred. The authors have identified a consistent remarkable process of fuions, promoted by inversions or simple repeats present adjacent to the origins of insertions.

The authors interrogate proteins implicated in end joining and observed a strong reduction in fusions in the absence of LIG4, suggesting NHEJ. However, complex events were not dependent on LIG4 or LIG3. Although RAD52 loss did not affect fusion frequency, it did markedly diminish the complex events, suggesting a strand invasion mechanism. The authors further find that polq fibroblasts have a diminished complex fusion events, but that the size of those that did occur increases, indicating that polq contributes to one pathway that creates complex insertions. Observations in a polq mutant background are nicely complemented by polq inhibitors that affect distinct polq activities. The authors nicely show that pold and pole contribute to insertions in complementary ways, perhaps explaining residual events that occur when polq is absent. The authors also use nocodazol to show that complex insertions occur in early mitosis. Overall, the authors present compelling evidence that mm-bir and polq are used in mitosis to create complex insertions at sites in the genome that have inversions and are prone to fork stalling when replication is perturbed.

We thank the reviewer for positive comments on our manuscript. We have provided further evidence to support a new model of mitotic MM-BIR, which involve MMEJ proteins initiating a Pol δ -dependent BIR pathway which is regulated by PIF1, POLD3 and PCNA. We have made extensive amendments to the manuscript to describe these new findings, which are highlighted in red on pages 12 and 13.

specific points:

1. For the 21q family of telomeres, could the authors provide an estimate of the number of telomeres involved.

We have included the following sentence in the manuscript (page 5).

we employed Transcription Activator-Like Effector Nucleases (TALENs) to generate DNA double-strand breaks (DSBs) about 1.3 kb from the telomeres of the 21q and 16p families of related telomeres (Fig. 2a), between them encompassing at least 19 chromosome ends.

2. It is curious that an entire family of telomeres would almost always experience terminal deletions prior to fusion. Why would this be? Is there a subtelomere sequence that might be responsible for this instability?

Terminal deletion is a characteristic of telomere fusion events, that we first observed at the XpYp and 17p telomeres (Capper et al. 2007). Extending our analysis to include multiple telomeres revealed a similar pattern (Letsolo et al. 2010). We considered that this was a function of error prone end joining mediated by MMEJ. However, 21q family telomeres appear to be subjected to larger deletion events (Fig. 1e). One possible reason could be that these telomeres are prone to R-loop formation, which may lead to increased DNA resection and deletion (Ngo et.al 2021).

Capper, R. et al. The nature of telomere fusion and a definition of the critical telomere length in human cells. *Genes Dev.* 21, 2495–2508 (2007)

Letsolo, B. T., Rowson, J. & Baird, D. M. Fusion of short telomeres in human cells is characterized by extensive deletion and microhomology, and can result in complex rearrangements. *Nucleic Acids Res* 38, 1841–52 (2010).

Ngo, G. H. P., Grimstead, J. W. & Baird, D. M. UPF1 promotes the formation of R loops to stimulate DNA double-strand break repair. *Nat Commun* 12, 3849 (2021).

3. For comparison with Talen data in Fig. 3b, it would be helpful if the authors could add a bar for the fraction of rearrangements that occur in crisis in the absence of telomerase. This would allow readers to readily assess the subtelomeric Talen DSB insertions with those that occur at telomeres that become critically short in the absence of telomerase.

Thank you for suggesting this. We have added a new figure for this useful comparison (Fig.2d, together with U2OS as requested by Reviewer 3).

4. Polq facilitates DSB repair in mitosis. To confirm this, the authors inactivated the G2/M DNA damage checkpoint (line 250). It would be helpful if the authors could explain here what happens to polq-healed DSBs if these inhibitors are used. Has polq activity been shown

to be inhibited by or increased if the the G2/M DNA damage checkpoint is inhibited? This would help the reader to understand the high levels of insertion that occur when these inhibitors are use, and the relationship of these inhibitors to mitosis and the mitotic repair activity of polq.

POLQ is activated in mitosis by PLK1-induced phosphorylation and CDK1/PLK1 induced phosphorylation of RHINO (Brambati et al 2023, 2023 Gelot et.al. 2023). We propose that G2/M checkpoint inhibition forces more cells to enter mitosis with unrepaired DNA double strand breaks (DSBs), resulting in more DSBs being repaired by the POLQ pathway. This is supported by our observation that the high level of insertions that occur in WEE1 inhibitor-treated cells can be reduced by two POLQ inhibitors (Supplementary Fig.8a).

To make this clearer, we have modified the sentences below (page 8-9).

Polθ facilitates the repair of DSBs in mitosis following activation by PLK1-induced phosphorylation and CDK1/PLK1-induced phosphorylation of RHINO³⁴⁻³⁶. To confirm whether telomere dysfunction-induced CCRs occur in mitosis, we examined the effect of forcing premature cellular entry into mitosis in the presence of DSBs, which would be expected to increase the number of DSBs being repaired by activated POLQ in mitosis. To do this, we inactivated the G2/M DNA damage checkpoint pathway regulated by ATR, CHK1 and WEE1³⁷.

Brambati, A. et al. RHINO directs MMEJ to repair DNA breaks in mitosis. *Science* 381, 653–660 (2023).

Gelot, C. et al. Poltheta is phosphorylated by PLK1 to repair double-strand breaks in mitosis. *Nature* 621, 415–422 (2023).

5. ‘template switching, and this process appear to continue until’; appears

We have corrected this typo (line 428).

Reviewer #3 (Remarks to the Author):

Comments on “Mitotic Microhomology-Mediated Break-Induced Replication Promotes Chromoanasythesis”

In this manuscript, the authors describe that upon telomere crisis, chromoanasythesis—a type of complex chromosome rearrangement (CCR) characterized by numerous insertions with microhomologies at tightly clustered breakpoints due to template switching—occurs via microhomology-mediated break-induced replication (MM-BIR). Utilizing Oxford nanopore sequencing technology for single-molecule long-read DNA sequencing, the authors analyze DNA and subtelomeric DNA. They report that chromoanasythesis takes place at short telomeres and sub-telomeric double-strand DNA breaks, asserting that MM-BIR occurs during mitosis and is promoted by RAD52, Polθ, and Polδ. This analysis could shed light on how chromosome rearrangements at the telomere region promote cancer and some congenital disorders. While I find this work interesting, I believe the data presented are premature to fully support the authors’ conclusions. Below are my comments.

Major Comments:

1. Complex Chromosomal Rearrangement Analysis: I acknowledge that analyzing complex chromosomal rearrangements is challenging. However, the authors claim that chromoanasythesis is the primary CCR responsible for telomere fusions and aberrations during replication stress, telomere crisis, and double-strand breaks at the telomere. The authors appear to have overlooked other CCR including chromothripsis. The analysis of LIG3/LIG4 abrogation—responsible for non-homologous end joining (NHEJ)—did not have a substantial impact. As is well known, chromoanasythesis can occur alongside chromothripsis. I am not fully convinced that chromothripsis, kataegis, or other types of CCR can be disregarded concerning telomere-telomere fusion under conditions of telomere crisis. The frequencies reported in the analyzed data do not adequately exclude other CCR mechanisms. Furthermore, in the discussion section, the authors suggest that replication stress triggers chromoanasythesis, while micronuclei or chromatin bridges serve as sources of chromothripsis. This idea is intriguing, but with the provided data, I require more convincing evidence.

Thank you for raising this important point. To clarify, we are NOT proposing that chromothripsis does not occur in telomere crisis. Chromothripsis clearly does occur in telomere crisis, which has been shown in seminal studies cited in our introduction and discussion. However, detection of chromothripsis is difficult in our CCR assay due to the limits of PCR amplification.

Instead, we are proposing a MM-BIR model to exclusively explain the occurrence of smaller scale CCRs like replication-based chromoanasythesis, which can be detected using our PCR sequencing assay. Our new MM-BIR model (see comment no.2 below) which incorporates proteins involved in MMEJ and BIR, does not completely rule out the involvement of MMEJ-induced ligation in the generation of CCRs. This could provide an explanation for the co-occurrence of chromoanasythesis and chromothripsis. We have clarified these issues in the new paragraph below (page 16).

Despite the involvement of mitotic MM-BIR in chromoanasythesis, our model does not completely rule out a role for MMEJ-induced ligation. Due to the presence of MMEJ factors Pol θ and Pol δ , it is possible that once DNA ends engaged in MM-BIR dissociate from their template, they can switch to an MMEJ mode and ligate with other DNA fragments. In support of this hypothesis, unique insertion patterns observed in WEE1 inhibitor treated cells show that short, fragmented DNA can be ligated into some CCR events (Supplementary Fig. 7c-f). Furthermore, most of the MM-BIR events detected here rely on MMEJ to ligate with another telomere and complete the repair process. The tight association between MM-BIR and MMEJ proteins could explain the exclusive requirement of LIG1/3, instead of LIG4, in MM-BIR. This plasticity between MM-BIR and MMEJ usage could contribute to co-occurrence of chromoanasythesis and chromothripsis^{8,13,14,17,21}. However, the detection of such events, especially megabase-scale DNA fragmentation and ligation described in chromothripsis is difficult in our assay due to the amplification limit of PCR.

2. Role of RAD52 in MM-BIR: The authors argue that MM-BIR is responsible for chromoanasythesis and assert that RAD52 is essential for this process. However, this contradicts established knowledge; two major forms of BIR have been documented: RAD52-dependent and RAD52-independent. Many studies indicate that MM-BIR does not require RAD52. Additionally, the data in this manuscript do not convincingly demonstrate that

RAD52 is critically required for the process. I encourage the authors to investigate this matter more thoroughly to ascertain whether the process is indeed MM-BIR and to exclude other forms of BIR.

We thank the reviewer for raising this point. To confirm the involvement of MM-BIR, we examined the requirement for PIF1 and POLD3, two processivity factors specifically required for extensive DNA synthesis during BIR. We generated PIF1 and POLD3 mutant cell lines and further knocked down these genes using siRNA. Importantly, PIF1 depletion reduced the size of complex insertions (Fig.6c), supporting the hypothesis that long insertions detected in our CCR assay are generated by BIR. This is further supported by our observation that depletion of PCNA, which is essential for PIF1 activity in BIR (Buzovetsky et al 2017, Li et al 2021), induced the same short insertion phenotype (Fig.6f). On the other hand, POLD3 depletion increased the number of complex insertions (Fig.6b), showing that these CCRs are regulated by the stability of Pol δ -dependent DNA replication, further supporting the involvement of BIR.

In response to comment no. 8 below, we have replotted our staggered graph to better present our data. The new graph (Fig.3b) clearly shows that RAD52 KO reduced CCRs, supporting its role in mitotic MM-BIR. This is consistent with its roles in initiating MiDAS and BIR in mitosis (Bhowmick et al 2016, Li et al 2021).

Based on these findings, we propose a new model of mitotic MM-BIR (Fig.7 and discussion), which involves MMEJ proteins initiating a Pol δ -dependent BIR pathway which is regulated by PIF1, POLD3 and PCNA. This pathway requires novel and unexpected collaboration between proteins involved in MMEJ (Pol θ) and BIR (PIF1, POLD3) in mitosis, which helps to explain its highly mutagenic nature.

We have made extensive amendments to the manuscript to describe these new findings, they are highlighted in red on pages 12 and 13.

Buzovetsky, O. et al. Role of the Pif1-PCNA Complex in Pol δ -Dependent Strand Displacement DNA Synthesis and Break-Induced Replication. *Cell Reports* 21, 1707–1714 (2017).

Li, S. et al. PIF1 helicase promotes break-induced replication in mammalian cells. *The EMBO Journal* 40, e104509 (2021).

Bhowmick, R., Minocherhomji, S. & Hickson, I. D. RAD52 Facilitates Mitotic DNA Synthesis Following Replication Stress. *Mol Cell* 64, 1117–1126 (2016).

3. Analysis of Deletion Size and Microhomology: The analysis regarding deletion sizes and microhomologies at the breakpoints does not sufficiently clarify the complexity of the CCR within telomere crisis. For instance, the presence of large deletions and a wide range in insertion sizes (from 55 bp to 3,916 bp) raises questions. Can the authors completely rule out the possibility of standard BIR in this context?

Our new MM-BIR model which incorporates non-processive (Pol θ) and processive DNA polymerases (Pol δ + POLD3/PIF1) in MM-BIR, distinguishes it from standard BIR and provides an explanation for the occurrence of both short and long insertions. In addition, high

levels of microhomology also sets it apart from standard BIR (see minor comments no.1 below).

We propose that the large deletions of telomere-adjacent DNA, especially at 21q family telomeres (Fig.1e), could be due to DNA resection (Ngo et al 2021), which could occur at these DNA ends prior to the initiation of MM-BIR. However, MM-BIR appears not to be affected by this process as it can be detected at the 21q family telomeres (Chr16 for example, Fig.1h) or other telomeres that are not subjected to such large deletions (Chr17 for example, Fig.1j).

Ngo, G. H. P., Grimstead, J. W. & Baird, D. M. UPF1 promotes the formation of R loops to stimulate DNA double-strand break repair. *Nat Commun* 12, 3849 (2021).

4. The authors utilized normal fibroblasts in culture that undergo telomere crisis after a finite number of cell divisions. Additionally, they employed Aphidicolin to induce replication stress. In this context, the telomere DNA damage appears to be ATR-dependent, which differs significantly from TALEN-induced double-strand breaks (DSBs). The employment of FSLR analysis to investigate subtelomere DSB-induced CCR raises questions about why the authors found the process to be ATR-independent and how two distinct types of DNA lesions provoke the same MM-BIR pathway, leading to chromoanasythesis.

To answer this question, we have examined the role of ATM in regulating CCR. Importantly, ATM inhibition increases the level of CCRs (Supplementary Fig.6e), similar to the result observed following WEE1 and CHK1 inhibition. Our data is consistent with ATM playing a dominant role in activating G2/M DNA damage checkpoint at DSBs (Jazayeri et. al 2005), further supporting the important role of this checkpoint pathway in suppressing CCRs.

ATM is mainly activated by DSBs, whereas ATR is activated by ssDNA (Blackford et. al 2017). At DSBs, ATM is first activated, and then ATR is activated following DNA resection (Jazayeri et. al 2005). Dysfunctional telomeres can activate ATM or ATR (Kibe et. al 2016).

We propose that MM-BIR can be initiated at DSBs without extensive DNA resection as the binding of MH requires limited regions of ssDNA. This is supported by our finding that a reduction of DNA resection following ATM inhibition does not reduce MM-BIR (Supplementary Fig. 6b,c,e). This could help explain why TALEN-induced DSBs or uncapped telomeres provoke the same MM-BIR pathway, regardless of whether they activate ATM or ATR. The important criteria for the initiation of MM-BIR are the presence of these DNA ends in mitosis. This could happen when DSBs occur/telomeres uncap in mitosis or when unrepaired DSBs or uncapped telomeres persist into mitosis.

Jazayeri, A. et al. ATM- and cell cycle-dependent regulation of ATR in response to DNA double-strand breaks. *Nat Cell Biol* 8, 37–45 (2006).

Blackford, A. N. & Jackson, S. P. ATM, ATR, and DNA-PK: The Trinity at the Heart of the DNA Damage Response. *Molecular Cell* 66, 801–817 (2017).

Kibe, T., Zimmermann, M. & de Lange, T. TPP1 Blocks an ATR-Mediated Resection Mechanism at Telomeres. *Molecular Cell* 61, 236–246 (2016).

We have amended the manuscript on page 9 with the following text:

To examine the role of ATM, we treated cells with an ATM inhibitor, KU-60019 and observed increased numbers of cells in G1, accompanied by a reduction of cells in G2/M, one day following the induction of DSBs (Supplementary Fig. 6a). This was similar to cells treated with WEE1 or CHK1 inhibitors and suggest that more cells progressed through mitosis due to the absence of G2/M checkpoint. However, cells treated with ATM inhibitor continued to progress through the cell cycle, likely due to the absence of G1 checkpoint, and by day 2, there were more cells in G2/M compared to DMSO treated cells (Supplementary Fig. 6a). ATM inhibition strongly reduced fusions associated with deletion and increased blunt junctions (Supplementary Fig. 6b, 6c, 6d), likely due to its role in promoting DNA resection³⁸. Importantly, ATM inhibition increased the frequency of complex insertions, further supporting the role of G2/M checkpoint machinery in suppressing chromoanasythesis (Supplementary Fig. 6e).

5. MM-BIR During Mitosis: The authors assert that MM-BIR occurs during mitosis, linking it to mitotic DNA synthesis (MiDas), which is typically associated with BIR. However, the claim that ATR inhibitors had marginal effects while WEE1 and CHK1 inhibitors increased CCR raises questions. Additionally, the MiDas assay depicted in the figures does not explicitly involve telomeres, making it unclear whether the sequence analysis is relevant to the MiDas assay presented in this work. I recommend that the authors repeat experiments with the inhibitors and also employ gene depletion methods (such as siRNA, CRISPR, or shRNA) to corroborate the analysis concerning ATR, WEE1, CHK1, LIG3, LIG4, POLQ, etc. This is essential for reaching any firm conclusions.

We have found that ATM plays a dominant role at these DSBs in suppressing CCRs (see comment 4 above), which likely explain why ATR inhibitor has marginal effect on CCRs. Our data is further supported by a paper showing that ATR inhibitor does not induce premature mitotic entry as efficiently as WEE1 or CHK1 inhibitors (Mak et. al 2014).

To confirm the occurrence of MiDAS at telomeres, we have performed MiDAS in conjunction with telomere-FISH and showed that both telomeric and non-telomeric MiDAS were significantly reduced when POLQ is depleted. These results provide further evidence that POLQ promotes MM-BIR at both telomeric and non-telomeric loci (Fig.5d-f, supplementary Fig.9a).

We have made extensive amendments to the manuscript to describe these new findings, they are highlighted in red on pages 9 and 11.

Mak, J. P. Y., Man, W. Y., Ma, H. T. & Poon, R. Y. C. Pharmacological targeting the ATR–CHK1–WEE1 axis involves balancing cell growth stimulation and apoptosis. *Oncotarget* 5, 10546–10557 (2014).

6. ALT Mechanism and Analysis of ALT Cell Lines: Break-induced replication is a key mechanism for Alternative Lengthening of Telomeres (ALT). The authors have conducted their analyses using normal fibroblasts. What about the ALT cells? Many reports suggest that replication stress provokes ALT. Therefore, it would be interesting to compare the signatures of telomere sequences between normal fibroblasts undergoing telomere crisis and ALT cells. I suggest that the authors compare the frequencies of CCR, along with the insertions, deletions, and copy numbers at both the telomere and subtelomere, to provide a more comprehensive understanding of the process in the context of ALT.

Due to low level of CCRs observed in fibroblasts which make their analysis challenging (Fig.2d), we tried to answer these questions by performing TALEN-induced telomere fusion assay in HCT116 (telomerase-positive) versus U2OS (ALT-positive) cell lines. We found that U2OS cells have a lower level of simple insertions and complex insertions/CCRs (Fig.2d), suggesting that ALT cells may have reduced level of NHEJ (simple insertion) and MM-BIR (complex insertion). While this is an interesting observation, we think that a thorough investigation of ALT in chromoanasythesis is outside the scope of this manuscript.

Minor Comments:

1. On page 6, line 166, the authors state that the microhomologies at the junctions were approximately 51.8%. What is the significance of the remaining percentage? Is this sufficient evidence to support the claim of MM-BIR?

Our analyses in HCT116 and RPE1 WT cells showed that 49.8-51.0% of the junctions contained overlap/microhomology (MH), 30.4-35.7% contained unmapped insertion/gap, and 14.5-18.6% were blunt (Fig.3d, supplementary Fig.5c). This high level of MH is consistent with a recent comparable study analysing POLQ-mediated repair which shows that 65-70% of junctions contain MH (Carvajal-Garcia et.al 2020, Fig.4F). The lower level of MH observed in our study could be due to unmapped insertions/gaps that cannot be mapped accurately due to their small size (<15bp).

Carvajal-Garcia, J. et al. Mechanistic basis for microhomology identification and genome scarring by polymerase theta. Proc Natl Acad Sci U S A 117, 8476–8485 (2020).

2. On page 7, line 200, and in Fig. 3A, the authors report that RAD52 knockout (KO) “mildly” reduced the level of insertions. I recommend a more precise characterization of the observed effects.

Thank you for making this suggestion. We have quantified the level of fusion for this figure (Supplementary Fig.4a) and rewritten these sentences (page 7).

Consistent with previous findings, a LIG4 knockout (KO) had a larger impact in reducing insertions compared to LIG3 KO (Fig. 3a, Supplementary Fig. 4a)²⁶. LIG3:LIG4 double KOs behaved similarly to LIG4 KO, whereas RAD52 KO reduced insertions weakly compared to LIG4 KO (Fig. 3a, Supplementary Fig. 4a).

In addition, we have also quantified the level of fusion in other important telomere fusion figures (supplementary Figs.5a, 6c, 9c, 10d).

3. On page 9, line 274, the reference to the TMEM105 oncogene being implicated in breast cancer appears to be over-interpreted. I suggest a more cautious phrasing to avoid overstating the evidence.

We have changed this sentence as written below (line 298).

Interestingly, this locus encodes TMEM105, a putative oncogene.

4. PolQ is not specific to the mitotic phase. I recommend toning down any references to it being associated exclusively with the mitotic cell cycle.

We have removed the sentences saying that PolQ acts exclusively in mitosis (line 259 and line 437).

5. On page 11, lines 337-339, the statement that “MM-BIR occurs in early mitosis to generate telomere dysfunction-induced chromoanasythesis independently of the formation of chromatin bridges or micronuclei” may be overstated, given that fibroblasts have a finite cell division cycle. Micro-nuclei result from chromosomal mis-segregation and a small number of micronuclei that do not degenerate can incorporate into the genome, leading to chromothripsis. It is unclear whether the experimental design sufficiently accounts for the number of cell divisions required for meaningful chromosomal mis-segregation. I am not sure whether the authors can definitively rule out the possibility of micronuclei formation and chromothripsis upon telomere crisis.

Again, we would like to emphasise that we are NOT proposing that chromothripsis do not occur in telomere crisis, and we are NOT saying that chromatin bridges or micronuclei do not contribute to CCRs during telomere crisis (see major point 1 above).

We are merely proposing that MM-BIR can occur independently of the formation of chromatin bridges or micronuclei, because we did not observe a difference in MM-BIR in the presence of nocodazole, which prevent entry into anaphase where bridges and micronuclei form (Fig.6h).

To better clarify this point, we have modified the following paragraph in discussion (page 16).

Currently, micronuclei and chromatin bridges represent two known sources of CCR, especially chromothripsis^{1,58}. We found that mitotic MM-BIR can generate CCRs in early mitosis independently of these two processes, thus revealing a novel source of CCRs. However, we could not rule out the involvement of these two processes in further amplifying CCRs together with MM-BIR.

6. Could the authors provide information on how many times the fibroblasts divided before committing to telomere crisis?

In addition to the growth curves provided in Supplementary Fig. 1a, we have added this information in the sentence below (page 3).

With this new methodology, we were able to specifically sequence telomere fusion PCR products isolated from four primary human fibroblast cell lines (WI-38, MRC5, HCA2 and IMR90) undergoing a telomere-driven replicative crisis after 16.89 (WI-38), 29.27 (MRC5), 29.22 (HCA2) and 35.75 (IMR90) of population doubling from the point of senescence (Fig. 1a, 1b, Supplementary Fig. 1a, 1b).

7. In Fig. 2, the authors conducted subtelomere analysis using HCT116 cells. Could they justify the use of different cell types in this context?

The level of CCRs is a lot lower in fibroblasts compared to HCT116 (Fig.2d), and it takes months for CCRs to accumulate in fibroblast, which make them challenging to study and compare between different mutants. This is why we used HCT116 and RPE1 cell lines with defined genetic modification, where CCRs can be generated in 2 days.

8. In Fig. 3B, the differences in CCR between LIG3, LIG4, and RAD52 are not as pronounced as the authors claim. A clearer presentation of the data may be helpful.

Thank you for suggesting this. We have changed the way we present the data to make it easier to compare the differences in CCRs (Fig.2d, 3b, 4b, 4e, 4i, 5h, 6b, 6e, 6h and supplementary fig. 1e, 6e, 8a, 9d).

In addition, we have also converted all staggered plots to bar graphs which are more easily interpretable.

9. In Fig. 4J, the MiDas assay with EdU should ideally be performed in conjunction with telomere-FISH (T-FISH) for enhanced clarity. Additionally, the differences noted in the results are not markedly clear.

We have performed MiDAS in conjunction with telomere-FISH and showed that both telomeric and non-telomeric MiDAS were significantly reduced when POLQ is inhibited (Fig.5d-f, supplementary Fig.9a, page 11).

REVIEWER COMMENTS

Reviewer #1 (Remarks to the Author):

The impressively revised manuscript contains a substantial number of new experiments. Several of these address my initial concerns while their outcomes strengthen the manuscript, as well as increase the impact/novelty of the study. I also felt that the discussion section has improved.

I therefore support publication of the manuscript in its current form.

(and I wish to compliment the authors on such nice work!)

We thank Reviewer 1 for positive comments on our manuscript.

Reviewer #2 (Remarks to the Author):

This is an outstanding manuscript that thoroughly interrogates the roles of RAD52, DNA pol theta, PIF1 PCNA and POLD in creation of complex fusion events that occur during early mitosis at shortened telomeres or at subtelomeric DSBs. These data are highly relevant to common genome rearrangements that occur during tumor development but are difficult to detect based on short read sequencing of tumor genomes. The authors have done a nice job of addressing concerns raised by the reviewers, including a number of experiments. This well written manuscript is suitable for publication in Nature Communications.

Reviewer #2 (Remarks on code availability):

Readme files are available that clearly indicate how to process their long read sequencing data sets.

We thank Reviewer 2 for encouraging comments on our manuscript.

Reviewer #3 (Remarks to the Author):

Comments for the revised manuscript.

Overall, the authors have adequately addressed several points raised by previous reviewers; however, there remain substantial conceptual and experimental gaps that significantly weaken the central claims of the manuscript. In its current form, despite the revision, the evidence provided is still insufficient. Although the study presents interesting observations and the topic is of high relevance, the current data does not adequately support the major conclusions of the manuscript. In my view, significant additional clarification and experimental strengthening are required before the study can be considered for publication. My major concerns are summarized below.

Major Concerns

1. Lack of direct cellular evidence demonstrating telomere crisis or telomere damage
The authors state that telomere crisis was induced either via TALEN mediated targeted telomere cutting or population doubling. However, no direct cellular or cytogenetic validation is presented to demonstrate that the cells indeed underwent telomere crisis or exhibited telomere dysfunction. To substantiate this foundational assumption, the manuscript should provide clear cellular evidence such as telomere dysfunction induced foci (TIF assay: γ H2AX or 53BP1 co localization with telomere FISH), metaphase spread analysis of chromosome end to end fusions, or telomere fragility markers. Without these data, it is difficult to interpret subsequent claims regarding telomere crisis induced genome rearrangements.

We thank Reviewer 3 for these suggestions. Telomere crisis is considered to be triggered when telomeres become critically short or dysfunctional and undergo fusion with other telomeres driving widespread genome instability (Capper et al., 2007). Therefore, we consider the detection of telomere fusions in our single-primer PCR assay to be strong evidence of telomere crisis. Moreover, the extended replicative lifespan and growth curves of these cultures (Supplementary Figure 1a) are consistent with these cells undergoing crisis as originally described by Counter et al (1992).

To provide additional validation of telomere crisis, we performed telomere dysfunction induced foci (TIF) assay (proximity ligation assay detecting colocalization of γ H2AX and TRF1) and metaphase spread analyses as suggested. These new experiments revealed TIF and chromosome end-to-end fusions in cells experiencing telomere crisis or TALEN-induced DSBs (Supplementary Figures. 1c, 1d, 2g, 2h), which provide direct cellular and cytogenetic evidence demonstrating telomere crisis/damage in our experimental system.

We have modified the following sentences to describe these new results.

With this new methodology, we were able to specifically sequence telomere fusion PCR products isolated from four primary human fibroblast cell lines (WI-38, MRC5, HCA2 and IMR90) undergoing a telomere-driven replicative crisis after 16.89 (WI-38), 29.27 (MRC5), 29.22 (HCA2) and 35.75 (IMR90) of population doubling from the point of senescence (Fig. 1a, 1b, Supplementary Fig. 1a, 1b, 1c, 1d). (page 3)

Consistent with our previous findings^{26,27}, we observed high frequencies of sub-telomeric DSBs and telomere fusions (Supplementary Fig. 2g, 2h) one day after transfection of TALEN-expressing plasmids into HCT116 wild-type (WT) cells. (page 5)

Capper, R. et al. The nature of telomere fusion and a definition of the critical telomere length in human cells. *Genes Dev* 21, 2495–508 (2007).

Counter, C. M. et al. Telomere shortening associated with chromosome instability is arrested in immortal cells which express telomerase activity. *EMBO J* 11, 1921–1929 (1992).

2. Insufficient explanation regarding cell type specific effects, particularly the stronger impact in HCT116

The manuscript does not adequately explain why chromoanasythesis is most pronounced in HCT116 cells while being minimal in U2OS, despite U2OS being an ALT cell line with high recombination activity, persistent replication stress, and MM-BIR competence. Given that ALT cells typically show elevated RAD52 and POLQ dependent recombination and robust templated insertion at DSBs, the unusually low insertion frequency observed in U2OS is unexpected. This discrepancy raises the possibility that the subtelomeric TALEN cut is inefficient in U2OS or that ALT cells process subtelomeric breaks differently from HCT116. To clarify this issue, the authors should examine TALEN cutting efficiency, end resection at the break, expression or activity of MM-BIR related factors (e.g., POLQ, RAD52), and ALT specific telomere features such as APBs or telomere length heterogeneity. Without addressing these possibilities, the cell type specific differences remain unexplained, weakening the mechanistic interpretation of the results.

Min, J. et al. Mechanisms of insertions at a DNA double-strand break. *Molecular Cell* **83**, 2434–2448 (2023).

We do not believe that the difference in complex insertion levels observed in U2OS cells is due to differences in TALEN cutting efficiency. A reduction in cutting efficiency would be expected to decrease all types of telomere fusion events, rather than selectively reducing insertion-containing fusions while increasing fusions without insertion, as observed in U2OS cells (Fig. 2d). Instead, our results suggest that ALT cells may process subtelomeric double-strand breaks differently from HCT116 cells, consistent with Reviewer 3's suggestion.

ALT cells are known to exhibit high levels of break-induced replication (BIR) activity and can undergo templated insertions at double-strand breaks (Min et al., 2023, as cited by Reviewer 3). However, the templated insertions characterised by Min et al. are POLQ-independent and are therefore mechanistically distinct from the insertions generated by the mitotic MM-BIR pathway first described in this manuscript. Consequently, there is no basis to suggest that MM-BIR would be more active in ALT cells. On the contrary, increased BIR activity in ALT cells would be expected to inhibit MM-BIR, as we show that these pathways are distinct and may potentially compete with one another. Thus, the observed reduction in MM-BIR in ALT cells actually supports our model.

We agree that a detailed analysis of how ALT cells regulate MM-BIR would be of interest; however, such an investigation is beyond the scope of the present study.

Min, J. et al. Mechanisms of insertions at a DNA double-strand break. *Molecular Cell* **83**, 2434–2448 (2023).

3. The evidence supporting mitotic MM-BIR relies almost exclusively on MiDAS data, which are of insufficient quality

The central claim that chromoanasythesis occurs during mitosis through mitotic MM-BIR is largely based on MiDAS. However, the MiDAS imaging data are of poor quality; EdU and telomere staining appear weak or inconsistent even in control samples, making interpretation unreliable.

We respectfully disagree with Reviewer 3's comments regarding the quality of the MiDAS images. Overall, the quality of these images is comparable to those we have recently published in *Nature Communications* and *Nature Structural and Molecular Biology* (Barwacz et al., 2025; Wu et al., 2023; Wu et al., 2020).

We would also like to draw Reviewer 3's attention to the fact that we and other groups have observed that telomere staining intensity can vary between individual chromosome ends in ALT cells, reflecting the heterogeneity in telomere length. We take this variability into account during data analysis by examining zoomed images on large, high-resolution computer screens. In addition, to ensure telomere signals could be seen easily in PDF files, we have adjusted the intensity of the green channel in the new Figure 5e. This simple change allows clear display of telomere and Edu foci and supports our view of 'the quality of these images is comparable to those we have recently published'.

Barwacz, S. A. et al. DNA double-strand break end resection factors and WRN facilitate mitotic DNA synthesis in human cells. *Nat Commun* 16, 7901 (2025).

Wu, W. et al. Mitotic DNA synthesis in response to replication stress requires the sequential action of DNA polymerases zeta and delta in human cells. *Nat Commun* 14, 706 (2023).

Wu, W. et al. RTEL1 suppresses G-quadruplex-associated R-loops at difficult-to-replicate loci in the human genome. *Nat Struct Mol Biol* 27, 424–437 (2020).

4. Unclear mechanistic relationship between RAD52 and Pol θ

The manuscript continues to lack conceptual clarity regarding the relationship between RAD52 and Pol θ in promoting chromoanagenesis. The authors simultaneously state that RAD52 is required while Pol θ acts in a RAD52 independent manner, which appears contradictory without mechanistic reconciliation. It is critical to clarify whether RAD52 functions upstream for ssDNA annealing/template preparation, while Pol θ drives later template switching and extension, or whether they act in parallel pathways. Without a clearer mechanistic explanation, the molecular model remains incomplete.

As shown in the first round of revision, RAD52 and POLQ act in parallel pathways to promote MM-BIR (Fig. 4h–j; Fig. 7). We propose that both proteins can initiate MM-BIR by promoting the annealing of microhomologies (Kent et al., 2015; Liang et al., 2024) and by stimulating the recruitment of Pol δ (Bhowmick et al., 2016; Stroik et al., 2023).

We have revised the sentences below to clarify this model (page 14).

In a mitotic MM-BIR model (Fig. 7), DNA ends utilise the strand annealing activities of RAD52 or Pol θ ^{59–61} to bind to exposed DNA with microhomologies in mitosis (Fig. 7a) and initiate end bridging DNA synthesis by Pol θ (Fig. 7b). The DNA synthesis initiated is short but becomes more processive following the recruitment of Pol δ by Pol θ or RAD52^{46,52}, and the arrival of processivity factors POLD3, PIF1 and PCNA (Fig. 7c).

Kent, T., Chandramouly, G., McDevitt, S. M., Ozdemir, A. Y. & Pomerantz, R. T. Mechanism of microhomology-mediated end-joining promoted by human DNA polymerase theta. *Nat Struct Mol Biol* 22, 230–7 (2015).

Liang, C.-C. et al. Mechanism of single-stranded DNA annealing by RAD52–RPA complex. *Nature* 629, 697–703 (2024).

Bhowmick, R., Minocherhomji, S. & Hickson, I. D. RAD52 Facilitates Mitotic DNA Synthesis Following Replication Stress. *Mol Cell* **64**, 1117–1126 (2016).

Stroik, S. et al. Stepwise requirements for polymerases delta and theta in theta-mediated end joining. *Nature* **623**, 836–841 (2023).

5. Interpretation of the POLD3 phenotype

Prior work has demonstrated that POLD3 loss induces substantial replication stress and cell death phenotypes. In your dataset, POLD3 depletion increases the frequency and tract length of complex insertions, which is interpreted as enhanced MM-BIR driven template switching. However, because POLD3 deficiency itself triggers pronounced replication stress, an alternative explanation is that the increase in complex events reflects globally elevated fork instability, rather than a direct mechanistic shift in MM-BIR dynamics. To distinguish between these possibilities, I strongly recommend performing POLD3 depletion under conditions where POLQ is simultaneously depleted, so that MM-BIR patching is suppressed but replication stress is still induced. This would clarify whether the increase in complex insertions is truly due to MM-BIR dysregulation or secondary to replication stress.

Buzovetsky, O. et al. Role of the Pif1-PCNA Complex in Pol δ -Dependent Strand Displacement DNA Synthesis and Break-Induced Replication. *Cell Reports* **21**, 1707–1714 (2017).

We thank the reviewer for raising this question. We performed additional experiments to examine the effect of POLD3 depletion in cells lacking POLQ (Supplementary Figs. 10g and 10h) and found that POLD3 depletion did not increase complex insertions in the POLQ knockout background (Supplementary Fig. 10i). These results suggest that the effect of POLD3 on complex insertion is mediated through POLQ-dependent MM-BIR rather than being a consequence of replication stress.

We have added the following sentence to describe this new finding (page 13)

Interestingly, POLD3 depletion did not increase complex insertion in the absence of POLQ (Supplementary Fig.10g, 10h, 10i), suggesting that frequent template switching observed following POLD3 depletion is due to Pol θ .

6. Interpretation of the PIF1 phenotype

Several studies have shown that PIF1 contributes to both replication coupled and replication independent forms of BIR. Therefore, the finding that complex insertions in your system are largely unaffected by PIF1 depletion is unexpected. This discrepancy raises the possibility that the events classified as complex insertions may not arise from a single, uniform recombination-based mechanism. Moreover, although LIG4 knockout did not produce a statistically significant reduction in complex insertions, there is a consistent trend toward decreased complex events in both the WT and LIG3 deficient backgrounds. Even if LIG4 is not directly required for complex insertions, there remains the possibility of MMEJ or SSA

mediated cut and paste like insertion events, which rely on LIG3 or LIG3/RAD52, respectively. Therefore, to more rigorously define the recombination pathway responsible for complex insertions, it would be informative to examine complex insertion frequency under conditions where both POLD3 and POLQ are simultaneously depleted. Dual suppression of POLD3 and POLQ mediated BIR synthesis should more completely eliminate MM-BIR activity.

Li, S. et al. PIF1 helicase promotes break-induced replication in mammalian cells. *EMBO Journal* 40, e104509 (2021).

Wu, T. et al. Break-induced replication is activated to repair R-loop-associated double-strand breaks in SETX-deficient cells. *Cell Reports* 44, (2025).

We have shown that Pif1 depletion affects MM-BIR by reducing the size of complex insertion events (Fig. 6c), which is consistent with our model that PIF1 supports the processivity of MM-BIR. This conclusion is further supported by our PCNA experiment (Fig. 6f). However, we speculate that PIF1 may play a weaker role in suppressing POLQ-mediated template switching, as it acts at a later stage of MM-BIR, following the full establishment of processive replication.

We also performed experiments to examine complex insertion frequency under conditions in which both POLD3 and POLQ were simultaneously depleted, as suggested by the reviewer. We found that POLD3 depletion did not affect complex insertion in the POLQ knockout background (Supplementary Fig. 10i; please see response to Question 5 above). This result is consistent with our hypothesis that POLD3 acts in the same pathway as POLQ to promote MM-BIR (Fig. 7). The low level of insertions observed in this background may be attributable to Pole activity (Fig. 5i).

7. ATR inhibition and cell cycle control

The manuscript concludes that ATR inhibition does not significantly alter cell cycle distribution. However, ATR inhibition causes a marked increase in simple insertions, consistent with the elevated simple insertion possibly by globally elevated DNA fragment pools that are captured by LIG4 dependent NHEJ. If ATR inhibition indeed produces widespread fork collapse and DNA fragmentation, one would mechanistically expect cell cycle perturbation, particularly at the G2/M checkpoint. It would strengthen the manuscript to clarify why ATR inhibition fails to affect cell cycle profiles despite an increase in simple insertion events.

In the first round of revision, we showed that ATM can compensate for ATR by stimulating checkpoint activation following subtelomeric double strand breaks induction (Supplementary Fig. 6), which may explain why ATR inhibition fails to affect the cell cycle profile. It is possible that ATR suppresses simple insertion through checkpoint-independent mechanisms. For example, ATR inhibition reduces DNA end resection and increases non-homologous end joining (NHEJ) (Dibitetto et al., 2020), which would be expected to increase NHEJ-mediated simple insertion events. ATR also has multiple additional checkpoint-independent functions that may contribute to this phenotype (Joo et al., 2024).

However, we believe that a detailed analysis of the checkpoint-independent roles of ATR in suppressing NHEJ-mediated simple insertion is beyond the scope of this study.

Dibitetto, D. et al. Intrinsic ATR signaling shapes DNA end resection and suppresses toxic DNA-PKcs signaling. *NAR Cancer* 2, zcaa006 (2020).

Joo, Y. K., Ramirez, C. & Kabeche, L. A TRilogy of ATR's Non-Canonical Roles Throughout the Cell Cycle and Its Relation to Cancer. *Cancers* 16, 3536 (2024).

REVIEWERS' COMMENTS

Reviewer #3 (Remarks to the Author):

The authors have adequately addressed the major concerns raised during the previous round of review, including those from junior scientists. The revised manuscript shows substantial improvement in both clarity and experimental support. In particular:

The authors now provide appropriate telomere dysfunction evidence using proximity ligation-based TIF assays and metaphase spread analyses, confirming that the experimental system indeed induces telomere damage.

The role of POLD3 in a POLQ-deficient background is now appropriately evaluated. Notably, POLD3 depletion does not increase complex insertions in the absence of POLQ, suggesting that the frequent template switching observed upon POLD3 depletion is POLQ-dependent.

The revisions also resolve earlier conceptual ambiguities. The authors now propose that RAD52 and POLQ operate in parallel, promoting microhomology annealing and POLD recruitment, respectively. Relevant textual sections have been appropriately edited to reflect this clarification.

This reviewer had requested additional experiments concerning TALEN cutting efficiency and the effects of ATR inhibition on cell-cycle profiles. While these were not addressed with new data, the authors have discussed these points in a manner that is reasonable at the current stage.

Taken together, the manuscript is now substantially strengthened and appropriate for publication in Nature Communications.

We thank the reviewer for positive comments on our manuscript.